# Audio, video, chat, email, or survey: How much does online interview mode matter?

**Maggie Oates\*, Kyle Crichton[ID], Lorrie Cranor[ID], Storm Budwig, Erica J. L. Weston, Brigette M. Bernagozzi, Julie Pagaduan**

Carnegie Mellon University, Pittsburgh, Pennsylvania, United States of America

\* moates@cmu.edu

## Abstract

In the design of qualitative interview studies, researchers are faced with the challenge of choosing between many different methods of interviewing participants. This decision is particularly important when sensitive topics are involved. Even prior to the Covid-19 pandemic, considerations of cost, logistics, and participant anonymity have increasingly pushed more interviews online. While previous work has anecdotally compared the advantages of different online interview methods, no empirical evaluation has been undertaken. To fill this gap, we conducted 154 interviews with sensitive questions across seven randomly assigned conditions, exploring differences arising from the mode (video, audio, email, instant chat, survey), anonymity level, and scheduling requirements. We surveyed interviewers and interviewees after their interview for perceptions on rapport, anonymity, and honesty. In addition, we completed a mock qualitative analysis, using the resulting codes as a measure of data equivalence. We note several qualitative differences across mode related to rapport, disclosure, and anonymity. However, we found little evidence to suggest that interview data was impacted by mode for outcomes related to interview experience or data equivalence. The most substantial differences were related logistics where we found substantially lower eligibility and completion rates, and higher time and monetary costs for audio and video modes.

## Introduction

In 2018, two of the authors conducted a pilot interview study to probe consumers' concerns when making online purchases of sensitive items such as sex aids, illegal drugs, or medical items. Fearing that participants would be wary to disclose their experiences to researchers, we focused on designing an online interview that minimized privacy concerns. We chose to eschew video in favor of instant messaging and invested substantial energy in a platform with strong security and privacy guarantees. After the study, we realized that despite our efforts against it, many participants had chosen to provide identifying information (e.g., a name or personal email address) and a few expressed difficulties with using the anonymous chat system. This experience led us to question whether our assumptions about mode and anonymity were aligned with practical priorities. After searching the literature, we were left without satisfying guidance on our questions about how to choose an online interview mode and how

**Data Availability Statement:** Research data cannot be shared publicly due to data-sharing restrictions present in the protocol approved by the CMU Institutional Review Board resulting from the

sensitive nature of the topics discussed in participant's interviews. The interview protocol contains questions that are highly sensitive in nature regarding topics such as death, sex, and relationships. Data collected during the interview from participants often contained detailed information about participant's past experiences. As such, the research team and the IRB agreed that the risk of re-identification, participant safety in participating in the study, and the potential mental health concerns related to the sensitive topics discussed in the interviews precluded, on an ethical level, the distribution of this data publicly even in an anonymized form. Aggregate data and results are provided in the manuscript and/or Supporting information files.

**Funding:** L.C. was named an Andrew Carnegie Fellow in 2019. The associated award, provided by the Carnegie Corporation of New York (https://www.carnegie.org/), helped to fund this study. The funders had no role in study design, data collection and analysis, decision to publish, or preparation of the manuscript.

**Competing interests:** The authors have declared that no competing interests exist.

much interviewees' anonymity and perceptions of anonymity were likely to impact interview results.

In the current literature, there are many guides outlining online interviewing methods [1, 2], most of which tend to provide general guidelines with a few examples from the authors' experience or case studies. While useful, their recommendations often rely on conjecture when discussing the topic of mode. For example, in their chapter "Remote Interviewing" [1, p. 95-96], King and Horrocks suggest that

> ... people may be more willing to disclose personal information online than they would be face to face (Joinson, 2005), probably due to the sense of anonymity they feel through communicating only by text. This tendency is likely to be present in any online interview technique, but the chatty, spontaneous style of IM [instant messaging] may be especially likely to facilitate openness from participants.

Many researchers in survey methods and communication theory have explored the impact of communication synchronicity, perceived anonymity, and other factors on interview-based research (see Related Work for examples and discussion). However, these studies are often conducted in specific contexts which limits broader applicability to qualitative interviewing generally. For example, while a health survey may be interested in eliminating desirability bias, a discursive qualitative researcher may be as interested in the retelling of a health event as hearing its "true" form. Similarly, while a lower completion rate across modes might cause concern for survey researchers concerned about self-selection bias, qualitative researchers are often focused on diverse or purposive sampling instead.

In comparing across different types of interviews, a substantial body of work has grown around online interviews and its use in qualitative research. However, these studies have focused exclusively on evaluating the effect of the online medium, comparing in-person to online interviews directly [1–7]. In contrast, the comparison between different types of online interviews has been left largely unexplored, leaving little guidance for researchers when deciding between different online platforms.

To address this gap, we conducted a mixed-methods study that randomized participants across seven different interview conditions. Our primary research question in this study is how do interview mode, scheduling requirements, and participant anonymity affect the experience of the interview and the data collected? As such, five of the conditions were used to evaluate differences across interview mode (video, audio, email, instant messaging, and survey). A sixth condition did not require the participant to schedule their interview, in this case a second survey condition, ahead of time and was used to evaluate the barrier posed by scheduling constraints. The seventh condition, a second chat interview, was designed to make the participant feel that they were not anonymous in order to assess how a participant's perception of anonymity affected their responses. Our dependent measures included qualitative data equivalence, rapport, completion rate, honesty, self-disclosure, and enjoyment. Surveys, which lack the conversational aspect and spontaneous followup questions valued in traditional interviews, are typically viewed as a distinct form of data collection. We chose to include surveys in our evaluation of mode to provide a baseline for comparison and to specifically assess the effects of scheduling. Acknowledging the inherent differences between these methods, most of our quantitative comparisons are done both with and without the survey conditions. In total, we conducted 154 online interviews between June and September 2019. Additional surveys were collected from both interviewees and interviewers after each interview and the transcripts were coded thematically to simulate a qualitative interview study.

In this study, we provide a detailed comparison between the common methods of conducting online interviews, which even prior to the Covid-19 pandemic had become increasingly used in qualitative research. With that in mind, we focus the interpretation of our results on interviews as a data collection method commonly used in qualitative research, rather than qualitative research methods more broadly. In this context, we find little evidence that interview mode affects the data collected and that the largest observed differences pertain to logistics, like eligibility and completion rates. **Our contributions** include: (1) the first systematic comparison of common online interview modes, (2) a new mixed-method approach to evaluate qualitative interview methods, and (3) practical advice for researchers designing online interviews.

## Related work

There is no one universal rubber-stamped method for conducting and analyzing qualitative interviews. However, best practices have been developed across a broad set of academic disciplines. These guidelines, which form the foundation of our experimental design, have been developed from a mix of theoretical grounding and practical experiential knowledge. In the following sections, we highlight related work pertaining to interview best-practices, experiments on survey modes and online communication psychology, and several empirical studies comparing interview modes.

### Desirable qualities of research interviews

King and Horrocks suggest three defining characteristics that make a good generic qualitative interview. First, the flexibility to deviate from the interview script and allow for open-ended follow up questions. Second, a focus on the interviewee's actual experience rather than their beliefs or opinions. Third, a positive relationship between the interviewer and the interviewee [1, p. 3].

Within these bounds, interviewers and methods researchers have constructed a diversity of (often overlapping) factors to help define what constitutes "actual experience" and evaluate the interviewer-interviewee relationship.

**Rapport and trust.** Intuitively, it is not surprising that many sources providing practical interview advice emphasize the importance of building rapport between the interviewer and interviewee. Rapport has been described as the "key ingredient of successful qualitative interviewing" [8]. Previous work has hypothesized that, in establishing a friendly and comforting relationship, an interviewee may be more motivated to participate and to provide accurate information during an interview [9]. However, researchers have also argued that high levels of rapport may motivate the interviewee to alter their responses to align with what they think the interviewer wants to hear or are generally more socially acceptable. This problem, also known as social desirability bias, is particularly challenging in interviews on sensitive topics where an interviewee's true behavior may not align socially defined norms [10].

Evaluating and measuring rapport as an operational concept has been criticized for its ambiguity and lack of structure [11]. Yet despite its murky underpinnings, rapport is widely recognized and discussed among interview researchers. Practical advice for building rapport includes negotiating clear expectations, dressing to an appropriate level of formality, mirroring terminology used by participants, and avoiding judgmental response [1]. Previous studies have measured rapport using a variety of methods. Hill and Hall (1963) evaluated rapport using post-interview surveys in which interviewer reported how *ill-at-ease* they felt during the interview and the degree to which the respondent was *favorable to the interview* [8]. Rapport has also been measured using eye contact [12], attentiveness [12, 13], interviewee motivation [1],

and empathetic moments [14]. Due to the multiple interview conditions evaluated in this study, most conducted without being able to see the interviewee, we opted for a self-reported measure based on Hill and Hall's questionnaire.

**Honesty.** In human subjects research, whether qualitative or quantitative, researchers face the fundamental challenge of accurately capturing how participants behave and think in the real world. This can be particularly challenging when using self-reported measures or interview responses, as the validity of the data collected is reliant on the honesty of the participant's responses. However, due to the lack of ground truth information, a participant's honesty is difficult to assess directly. This has been of particular interest to researchers who focus on sensitive behaviors like sexual activity and drug use. Fortunately, previous work in this area has found that self-reported measures of honesty are noisy but often correlated with actual behavior [15–17]. Therefore, we included a self-reported honesty measure in this study.

**Self-disclosure and intimacy.** While self-disclosure has many definitions, we prefer that of Moon (2000) which simply defines it as "any personal information that a person communicates to another" [18]. Conceptually, self-disclosure is closely related to the intimacy of a conversation. To measure this in an interview, self-disclosure is often separated into breadth, the quantity of different topics of information shared, and depth, the amount of personal detail shared in each topic [19]. Measures of self-disclosure have included counting the number of topics volunteered by participants [19], using a survey instrument [20], asking participants to self-rate their level of disclosure, asking conversation partners to rate each other, and using independent coders [20, 21]. In this study we employ two of these methods, one that leverages a previously validated instrument and another that applied qualitative coding to interview transcripts.

## Other factors that affect interview experience

While the purpose of this study is to measure the effects of interview mode, previous literature has identified several other factors that impact the interview experience and, as a result, the data that is collected.

**Interviewer effects.** Racism and race [22, 23], gender, age, and other identity factors can affect interview data and the interviewer-interviewee relationship [24, 25]. Several studies speculate that online modes with fewer visual cues (e.g. email) can minimize these effects. This is supported by evidence that interviewer effects are smaller in telephone interviews than face-to-face ones [3]. While we aimed to capture interview affects across mode, we also wanted to minimize this effect within mode. Therefore, our three interviewers were all of similar age and were of the same race and gender.

**Sensitive content.** In their review of previous studies involving surveys on sensitive topics, Tourangeau and Yan (2007) found that the inclusion of sensitive topics reduced the response rate in some surveys [26]. Sensitive topics have introduced additional challenges for research teams including intrusiveness, the threat of disclosure, potential consequences of disclosure such as legal sanctions, and social desirability bias [6]. Nandi and Platt (2017), in their study comparing face-to-face and telephone questionnaires, did not observe any mode-related differences except in response to a sensitive question on political identity [27]. This suggests that sensitive questions may heighten differences observed across mode. Since the interview questions in this study focused on sensitive topics, the research team took several steps to protect interviewees and interviewers. It was made clear to the interviewees in the beginning of the interview that they could skip any questions they felt uncomfortable with and at any point during or after the interview they could rescind their participation in the study. The study protocol included a process to follow in cases where interviewees had strong negative reactions to the

interview and ways to escalate any problems encountered during or after. The protocol also included contingency plans for cases where the interviewer felt uncomfortable or unsafe including the ability to end the interview immediately at their discretion.

**Social desirability bias.**   Social desirability bias is a phenomenon where interviewees alter or withhold their response because they are aware of the possible social implications of their intended answer. Social desirability bias has been interpreted by researchers as something that is inherent in the participant, possibly a part of their personality, or inherent to the topic of the interview question, as some behaviors and preferences are socially less accepted [28]. While the factor is difficult to measure directly, it is closely related to rapport, honesty, and self-disclosure. In our research design, we attempted to limit the opportunity for social desirability bias by randomizing interviewers across mode and collecting self-reported measures of honesty and rapport to validate our results.

**Perceived anonymity.**   Often, interactions conducted in an online setting provide users with greater opportunities for anonymity. Previous work has hypothesized that this factor creates opportunities for greater self-disclosure than in face-to-face conversation. Suler (2004) hypothesized that there exists an online disinhibition effect that allows individuals to express different aspects of themselves online that they would not in person [29]. Supporting this theory through empirical evidence, Joinson (2001) concluded that when people feel less identifiable, they tend to reveal more information about themselves and are less influenced by social desirability bias [30]. However, this effect may be context specific as Hollenbaugh and Evereett (2013) found an opposite relationship between anonymity and disclosure in their study of personal blogs online [31].

To measure perceived anonymity, Hite et al. (2014) developed and validated a five-question, context-independent measurement instrument to capture how users perceived their own anonymity [32]. In this study we leverage this instrument to assess how anonymous users felt across conditions.

## Evaluating online communication

Research in the fields of information processing and communication offer many useful paradigms to assess interviews in an online context. However, taken as a whole, these disciplines provide ambiguous or even contradictory recommendations. Media Richness Theory (MRT) proposes that "media richness" should be a key factor in choosing interview mode. The theory further claims that face-to-face communication is richer than digital communication and online modes that more closely resemble in-person communication, like video conferencing, are better suited for complex tasks like interviews [33]. However, a recent empirical study comparing audio and video communication online found that groups were more successful in collaboratively solving problems using audio conferencing. The study observed that groups interacted more cohesively, speaking out of turn less frequently, when there was a lack of video cues. As a result, teams were more successful in solving group problems, an ability known as Collective Intelligence (CI) [34]. In addition, Dennis et al. (2008), in their expansion on Media Richness Theory called Media Synchronicity Theory, push back on the idea that media richness is the ultimate quality for communication, concluding that there is no single media that is best for all tasks. They point out that that asynchronous modes are stronger for "conveyance processes" that convey or transmit information, and synchronous modes are better for "convergence processes" that find common meaning from information. [33]. While most research interviews are likely hybrid processes, they tend to have a greater focus on conveyance rather than convergence.

Other relevant theoretical models include the Social Information Processing theory (SIP) and the Social Identity Model of Deindividualization (SIDE). In their systematic review of studies that compared online and offline communication, Nguyen et al. (2012) applied these communication theories to predict whether self-disclosure varies by digital modes. However, they found that some theories did not clearly apply to digital modes and that some yielded contradictory predictions. Furthermore, in examining the literature as a whole, they found an inconclusive mix of studies reporting more, equal, and less self-disclosure for online modes as compared to offline [21]. In an earlier review of 18 experimental studies comparing group face-to-face communication with digital communications, Bordia (1997) found that conversations via digital communication tend to be longer, generate more ideas, have greater equity in participation, and reduce shared social pressures among participants. However, digital communication also lowered participant comprehension of the conversation [35].

## Comparing survey modes

A substantial body of work has been generated focusing on the effect that different methods of conducting surveys has on the quality of data collected. Several of these studies did not find any substantial differences across mode. Nandi and Platt (2017), in comparing social identity questions between face-to-face and telephone questionnaires, found "little evidence for specific mode effects." The only exception was for a sensitive question on political identity. They cautiously suggest this difference may be related to sensitivity, social desirability, or randomness [27]. Similarly, in comparing an online survey to an online instant chat questionnaire, Stieger (2006) found no differences in data quality [36].

Of the studies that did report differences across mode, two trends emerge. First, the effects that are found tend to be small. Second, no consensus arises between studies as to which modes are preferable [27]. One study that is particularly relevant to our work is that conducted by Tourangeau and Smith (1996) which collected information about sexual behavior using a digital survey, a digital survey that read questions aloud, and a face-to-face verbal survey where the interviewer recorded information digitally. They found that response rates did not differ across mode, but voluntary self-disclosure was higher in the digital surveys. However, their interpretation was that "computerization" itself had little impact on the level of reporting, but that the presence of an interviewer did [6].

## Comparing online interviews

Of the previous literature pertaining to online interviews, a large proportion has focused on comparing the online medium to traditional face-to-face methods. In general, these studies have found that online interviews provide several advantages over in-person interviews. This includes the ability to be geographically distant from participants, flexible scheduling, cost savings, and the mitigation of accessibility issues [2, 3, 7]. Common disadvantages include technical difficulties, lack of access to technology, digital privacy issues, and difficulties establishing rapport [1, 2, 4, 7]. Highlighting the double-edged nature of online modes, Weller (2017) observed that remote interviewing can increase disclosure, but can also lead to participants oversharing details, possibly to the point of harm [4].

Although online interviews have been typically conducted using audio or video chat platforms, email also provides an opportunity for remote interviewing. While previous work has suggested that email has fewer context clues to support clear communication and inhibits the building of rapport, it can also reduce interviewer effects and allow time for reflection. An additional concern unique to asynchronous interviews is that it is more common for interviewees to not respond and drop from the study in the middle of an interview, resulting in a

higher rate of incomplete interviews [3]. Comparing face-to-face and email interviews on breast feeding, a sensitive topic, Dowling (2012) noted several comparative benefits of using email. First, the asynchronous nature of email provided room for reflexivity and enabled "more thoughtful and considered responses." Second, email reduced the time required to conduct the interviews and the cost of transcription. Third, it alleviated the interviewing fatigue of the interviewer. Fourth, the researcher noted that "open and honest disclosure felt easier" in e-mail interviews. However, as previous literature has hypothesized, establishing rapport was more difficult and took more time [5].

More similar to our study, Jenner and Myers (2018) completed a mixed-methods examination of data equivalence, rapport, and disclosure between in-person and video interviews. They observed no differences in intimate self-disclosure, scheduling rates, or interview length between the two modes, but did observe less intimate disclosure for interviews held in public places. In one interview where a participant refused to share their video feed, effectively making it an audio interview, the researchers noted that the lack of nonverbal cues was challenging for the interviewer and resulted in a shorter interview. In particular, they note that it was difficult to assess whether "silence meant the participant was thinking about a response, confused by a question, or waiting for the interviewer to ask another question" [7].

While many studies have compared online and offline interview modes, or have evaluated the advantages or disadvantages of a specific online interview method, there has been no comprehensive empirical evaluation of online interview modes. This gap leaves researchers without clear guidance when designing online interview studies. As the trend towards online communication continues to grow, especially in light of the Covid-19 pandemic, the need to understand how different online modes impact communication only becomes more important. In this study, we contribute the first mixed-methods empirical comparison of different methods of conducting online interviews and provide insight into the effect of mode on recruitment, cost, rapport, honesty, self-disclosure, perceived anonymity, and qualitative data equivalence.

## Methods

Interviews are often used as part of qualitative and exploratory studies. However, to compare across online interview mode we employed an experimental approach that randomly assigned participants to conditions. This allows for a systematic comparison between conditions. Yet, we were wary of attempting to assess a qualitative method using quantitative experimental methods. As a result, we opted for an experimental, mixed-methods approach. In the following sections we provide an overview of the research study design, describe each of the seven experimental conditions, and outline the design of the interviews.

### Study design

To compare between different types of online interviews, we conducted 154 interviews between June and September 2019 in which participants were randomly assigned to one of the seven study conditions. Participants were recruited through Prolific Academic and screened using a short survey. We received 1,240 responses to our screening survey, of which 360 were eligible for participation, 337 consented to participate, and 154 ended up completing interviews. Eligible participants were then randomly assigned an interviewer and interview condition. After each interview completed, both the interviewee and interviewer completed a survey about their experience. Individuals who completed the screening survey were compensated $1 USD and those who completed the interview process were compensated $10 USD. An analysis team transcribed and coded all interview transcripts for themes to prepare the data for

analysis. This study received approval from the Institutional Review Board (IRB) at Carnegie Mellon University.

Wherever possible, we controlled for variables that are known to influence interviewer quality and experience, such as interviewer training and interviewer race (see Interviewers). At the same time, we were cognizant of the fact that one of the oft-cited benefits of qualitative interviewing is the capability of flexibility and spontaneous contextual adjustment [1, 2]. As a result, our study design decisions aimed for a precarious balance of ecological validity (e.g., are these interviews true to practical uses and qualitative methods?) and internal validity (e.g., can we make systematic comparisons across modes?).

A note on terminology, there were three teams involved in this study. The *interviewer team* and the *analysis team*, which included transcribers and coders, members were not aware of the high-level study design or research question. The *research team* members were aware of the study design. We sometimes abbreviate *interviewers* and *interviewees* as IER and IEE respectively.

## Experimental conditions

We assigned interviewees randomly to one of seven online interview conditions: video, audio, chat, non-anonymous chat, email, survey, and scheduled survey. The survey conditions were intended as a baseline to understand the effects that the presence of an interviewer would have. In these conditions, the lack of an interviewer precluded any interviewer-interviewee interactions and the possibility to ask followup questions. The anonymous and non-anonymous chat conditions served as as a test to evaluate the impact of anonymity. The survey and scheduled survey conditions were used to assess the cost of having to schedule an interview and the logistical overhead that imposed on participants. Table 1 provides a summary of the experimental conditions.

**Anonymity.** In all but one condition we discouraged, but did not prevent, participants from sharing identifying information for the study. This included encouraging the use of the anonymous email provided to participants by Prolific Academic for communication, discouraging users from entering their name when prompted by the video platform, and encouraging them to use pseudonyms during the interview. We instructed interviewers to avoid using the participant's names when addressing them in the interview. However, in the *Non-anonymous Chat* condition, we omitted all instructions discouraging interviewees from sharing their own identifying information. In addition, we instructed non-anonymous chat interviewers to use and repeat the participant's name if it was provided in order to emphasize to the participant that they were not anonymous.

**Scheduling.** All interviewers maintained a minimum of 20 hours/week available for interviewing. Eligible interviewees were invited to self-select an interview time slot from those

**Table 1. Summary of experimental conditions.**

| Condition | Emphasized Anonymity | Synchronous | Scheduled | *n* conducted |
|---|:---:|:---:|:---:|:---:|
| Video | ✓ | ✓ | ✓ | 19 |
| Audio | ✓ | ✓ | ✓ | 18 |
| Chat | ✓ | ✓ | ✓ | 23 |
| Non-anonymous Chat | x | ✓ | ✓ | 20 |
| Email | ✓ | x | ✓ | 26 |
| Survey | ✓ | N/A | x | 25 |
| Scheduled Survey | ✓ | N/A | ✓ | 23 |

available using the Calendly scheduling platform. To control for possible selection effects resulting from scheduling, we required all participants assigned to the *Email* and *Scheduled Survey* conditions to schedule the start of their interview.

## Interview design

The research team developed the interview script to balance a number of considerations. On one hand we reasoned that differences in self-disclosure, perceived anonymity, and honesty would be more pronounced across mode if the interviews focused on sensitive topics [27]. For interviews containing less sensitive content, we would expect mode to have a smaller effect. At the same time, we wanted to avoid topics that had a high likelihood of re-traumatizing participants, elicited health information, or elicited reports of illegal behavior. In addition, we wanted the interview to cover a variety of topics for increased generalizability. It is likely that interviews focused on a highly sensitive topic like sex result in a substantially different experience than interviews focused on something less intrusive like personal reflection. To provide continuity, we needed a plausible overarching research topic that united the questions and gave motivation to both interviewees and interviewers. The research team felt that his would allow interviewers to direct their questions and interviewees to provide answers in an organic way. Lastly, we aimed for an interview protocol that could be completed within 30–40 minutes.

To meet these goals, we selected a subset of questions from an interview study conducted by Moon et al. (2000) that examined online surveys on sensitive topics [18]. The subset of questions were selected to fit the above criteria, slightly modifying outdated language. Questions were roughly ordered by increasing sensitivity in order to allow the interviewee to get comfortable and build rapport with interviewer before getting into more the more difficult parts of the conversation. Interviewers were required to follow the interview script, but were encouraged to ask their own followup probes. The interview questions are summarized below and the full interview protocol is available in S1 Protocol.

1. What are your favorite things to do in your free time? (Free Time)

2. What characteristics of yourself are you most proud of? (Pride)

3. What are your feelings and attitudes about death? (Death)

4. What has been the biggest disappointment in your life? (Disappointment)

5. What is your most common sexual fantasy? (Sex)

6. What have you done in your life that you feel most guilty about? (Guilt)

7. What characteristics of your best friend really bother you? (Best Friend)

8. Is there anything you want to add? (Anything To Add)

**Interview logistics.**   We chose online platforms to use based on interviewee usability, privacy options, and a feature set that matched our condition requirements. For video, audio, and chat conditions, we used Zoom cloud meetings. For email conditions, participants were encouraged to use the anonymous email address and platform provided by Prolific Academic, but they could also choose to use a personal email address and platform of their choice. Survey conditions were hosted on Qualtrics.

Interview calendar invites were automatically sent via email immediately after sign-up and a reminder email was sent 24 hours before and 30 minutes before the interview start time. Reminders included instructions on how to access and test the assigned platform. In pilot

testing, we found that interviews usually lasted around 30 minutes. This included email, where we asked pilot testers to track the cumulative time they spent reading and answering study emails. We asked both interviewers and interviewees to allot at least 45 minutes per interview to allow ample time to complete the interview. Participants who completed an interview were compensated $10 USD.

**Interviewers.** Unlike many studies, our interviewers were considered research participants due to the fact that we analyzed their behavior in addition to interviewee behavior. For recruitment, we opened a screening survey for paid interviewers and distributed it across the departments at our university. Given that interviewer identity and presentation is a known variable in interview experience (See Interviewer Effects), we hoped to find a set of interviewers with similar demographics and levels of experience. All applicants were students at our university. From our pool, we selected three interviewers with similar traits: all three were White, women, aged 20–25 years, fluent English-speakers, willing to broach sensitive topics, with a similar gender presentation and hairstyles, and who had no previous experience with research interviewing. All interviewers received the same paid five hour training. The training curriculum included information on research ethics, safety, confidentiality, followup probes, and hands-on practice. Given the sensitive topic matter, interviewers were encouraged to stop an interview at any point if they felt uncomfortable, without loss of compensation. All interviewers completed a consent form and took the required human subjects training approved by the Institutional Review Board at Carnegie Mellon University.

During the study, each interviewer met weekly with a member of the research team to review progress, check compliance with the research protocol, debrief emotional concerns, and troubleshoot technical issues. Interviewers were compensated $60 USD for training, $12 USD per check-in meeting, $12 USD for each completed interview, and a 7% bonus on all earnings if they completed the entire study. One interviewer chose to leave the study early due to time constraints, so interviewees were reassigned to the remaining interviewers at random. As a result, Interviewer 1 and Interviewer 2 completed 42% and 36% of all interviews respectively, while Interviewer 3 completed only 22% of the interviews.

While interviewer effects are difficult to measure directly, we looked for differences in interviews conducted by different interviewers using several indirect measures. This included the interviewee's self-reported rapport rating with the interviewer, the interviewee's self-reported honesty, interview word count, qualitative code count, technical difficulties and completion rates. No substantial differences were found between interviewers across any of these metrics, indicating that interviewer effects were likely not a source of differences.

**Deception.** To mitigate confirmation bias, the research design included mild deception for both the interviewees and interviewers. This use of deception was approved by the Institutional Review Board at Carnegie Mellon University. We informed interviewees that the study broadly focused on improving online interviews. We informed interviewers that the study focused broadly on improving interviewer training. Real-world interviews require a research question to guide the arc of the interview, so we also informed interviewers, transcribers, and coders that the study specifically focused on how research participants narrativize sensitive memories, and encouraged interviewers to ask followup questions that elicited anecdotes and details. If the interviewees inquired about the study purpose, the interviewer was instructed to give a broad answer about improving online interviews, and direct further questions to the research team.

We debriefed interviewees in writing on the last page of their followup survey that they completed after their interview. This followup included information about how they would receive their compensation. We debriefed interviewers, transcribers, and coders in person when they concluded their portion of the study.

## Data collection

The data for this study was collected from multiple sources. This includes the interviewee screening survey, interview transcripts, and post-interview surveys. In addition, the transcripts were coded as if conducting a qualitative study to evaluate differences in data equivalence.

**Interviewee recruitment.** Interviewees were recruited using the Prolific Academic digital research platform. Participants were recruited from a pool of individuals who were 18 years of age or older and resided in the United States. A 5-minute screening survey was used to determine eligibility for the study and individuals who completed the survey were compensated $1 USD. A full version of the screening survey can be found in S4 Protocol. The screening survey included questions related to demographics, work habits, willingness to complete five different types of online interviews, and reliable access to the technology needed to participate in the interviews. Participants were deemed eligible if they were willing to complete all five modes of interviewing, were willing to be recorded, and had regular access to a private, quiet space.

Of 1,240 people screened, 27% were eligible to participate. The most common reason for ineligibility was being unwilling to participate in one or more of different interview modes (usually Video or Audio) and/or being unwilling to be recorded (see Recruitment and Logistics). Eligible participants were batched and randomly assigned an interviewer and interview mode. Interviewers sent a uniform interview invitation to all participants. If a participant did not schedule an interview within two weeks of the invitation, they were dropped from the active pool and replaced with a new randomly-assigned interviewee candidate.

We anticipated that different modes would yield different completion rates. Following reliability guidance from Simmons et al. (2011), we aimed to complete at least 20 interviews in each mode [37]. For scheduling and monetary reasons, we capped each condition at 25 participants. After having difficulty completing enough Video and Audio interviews, we extended the duration of our study for an additional month. Due to monetary and scheduling constraints, we chose to end interviews after this time period, yielding 18–26 interviews per condition. See Table 1 for the complete list of interviews conducted per mode.

**Post-interview survey.** After each completed interview, both the interviewee and interviewer were required to immediately complete a post-interview survey. For interviewers, this survey inquired about technical difficulties, interruptions, their discomfort, and their perception of the interviewee's comfort. It required less than three minutes to complete. A full version of the interviewer survey can be found in S5 Protocol. For interviewees, the post-interview survey focused on perceptions of the interview experience, the online platform used for the interview, and their opinions on their own disclosures. This survey took less than 15 minutes to complete and was slightly tailored by mode. For example, rating the audio quality was not relevant for all modes. Table 2 summarizes the interviewee followup survey, and a full version can be found in S4 Protocol.

**Interview transcription.** In order to compare the interview content on relatively even footing, interviews were all transcribed and reformatted uniformly. For audio and video interviews, Zoom's automatic transcription service was used to generate the initial text transcript. Then, a four-person team validated the transcription and removed personally identifiable information. In order to maintain possible non-textual communication, transcribers were asked to capture any changes in tone of voice, meaningful pauses, and body language used (e.g., *laughs* or *haltingly*). For the chat, email, and survey conditions, transcribers consolidated emails, exported timestamps, tagged the responses by question, and broke up long responses into shorter utterances. Other text-based communication vectors like emojis or spelling choices were retained in chat, email, and survey conditions. For all modes, the text transcripts were consolidated into a standard format. Coders were instructed to separately flag portions of

**Table 2. Summary of interviewee post-interview survey.**

| Topic | # of Qs | Details |
|---|---|---|
| Desirable responding | 16 | Hart's 2015 BIDR-16 index [38] to assess socially desirable responding |
| Comfort | 2 | feeling ill at ease, feeling motivated |
| Rapport | 5–7 | Questions from a medical interview rating scale [12] and a study on rapport [8] |
| Self-disclosure | 11 | self-reported self-disclosure level [20] |
| Perceived anonymity | 4 | Instrument for perceived anonymity [32] |
| Honesty | 4 | level of honesty and withholding details |
| Environment | 4 | possible distractions, noise level, feeling overheard |
| Online platform | 2 | familiarity and ease-of-use of the interview platform |
| Question enjoyment | 2 | Most enjoyable and most uncomfortable question |
| Open comments | 1 | *Is there anything else you'd like to share with us about your interview experience?* |

the interview that were related to the initial question and those that were followup questions. Each transcript was reviewed by a second transcriber to mitigate errors. Transcribers discussed any differences in interpretation and resolved them.

**Qualitative coding.** Previous studies evaluating qualitative interviews have used the number of "topics," generally defined, to measure the breadth of information disclosed during an interview [19, 21, 39]. This method is based on early work in offline communication and Social Penetration Theory [40]. However, the definition of topics in these studies depended on the context of the research study. Given the broad set of interview questions in our study, we decided to employ the common practice of qualitative coding, using emergent coding techniques, to define the set of topic categories [41]. To this end, a subset of questions were coded as if conducting an actual qualitative study. Four qualitative coders, two of whom were also on the transcription team, were split into pairs to code separate interview questions. All coders received the same introductory training on qualitative content coding, with an overview from Lazar et al's (2017) book chapter on "Analyzing Qualitative Data" and specific advice from Erlingsson and Brysiewicz's (2017) "A hands-on guide to doing content analysis" [42, 43]. Each question was coded using a main research question along with the overarching theme of how participant's narrativized their response. For example, when coding the interview question pertaining to the participant's views on death, coders operated under the motivating research question "what are Americans' feelings and attitudes about death?" The detailed coding handbook is available in S2 Protocol.

For each question that was evaluated using this method, a codebook was generated using the following process. First, each coder sampled approximately 250 text utterances and highlighted what Erlingsson and Brysiewicz refer to as "meaning units": a lower level abstraction of an interviewee's response [43]. Second, the two-person team met to reconcile and consolidate their work into "condensed meaning units." At this step, the original "meaning units" were shortened while retaining the response's core meaning and a common interpretation was agreed upon between coders. Third, the pair examined and grouped those units into codes, and then codes into higher-level categories. To avoid large codebooks, the research team instructed the coders to aim for a codebook that had 5–35 codes. Fourth, the coders drew another sample of 100–200 text utterances and individually applied the initial draft of the codebook to the new text. At this point, coders were instructed to note any points of ambiguity, utterances that did not fit well into existing codes, and points of particular interest. Fifth, the team met to compare their codes, deliberate on points of disagreement, and adjust the codebook accordingly. These steps were repeated as many times as needed, modifying the

codebook until both coders were in agreement or they thought that disagreements stemmed from subjectivity of the reader rather than ambiguity in the codebook. In practice, the pairs developed three to five different drafts of the codebook per question. In the final step, each coder individually re-coded the entire response set with the final codebook (≈150 interviews and ≈1500 text utterances). The pair met a final time to reconcile final codes. We emphasized to coders that differences in codes were not necessarily errors, they could also be the product of reasonable differences in interpretation or experience. As a result, the coders could agree to include both their codes. The coding teams did not use specific qualitative data analysis software. Instead, shared spreadsheet templates were used to generate the codebooks and code the responses.

To practice, each two-person coding team was assigned one question set to get hands-on experience with the coding process. These codebooks and codes were not used in our analysis. After the initial practice round, coding team 1 coded the questions related to guilt, best friends, and death. Coding team 2 coded pride and sex. Rather than do a full qualitative analysis of themes, we stopped our qualitative coding at the level of codes and categories. Our intention was to create data that could both summarize an interview as a qualitative researcher genuinely might *and* allow for a meaningful quantitative and qualitative comparison across interview modes. After all transcripts were coded, we divided the texts by mode in order to analyze possible differences in data equivalence.

## Edge cases

Throughout our analyses, there were minor variations in sample sizes due to anomalies in the protocol. In general, we included edge cases in the analysis except where they would bias results. Sub-sample sizes and exclusions are noted throughout where relevant.

**Partial interviews.**   Several interviews were only partially completed during the course of the study due to several factors. In five interviews (four chat, one email) the interviewer skipped a question. These interviews were considered complete and were included in the analysis done at the question level for those that were not skipped. These interviews were excluded from any analysis done on the content of the interview as a whole. For example, when computing average word count at the *question* level, we included interviews missing questions, but excluded them for word counts at the *interview* level (see Word Count). Notably, missing questions were distributed evenly across interviewers, but occurred only in text-based modes. This suggests that text modes were more prone to interviewer error. In three separate cases (two email, one chat) an interview was started but abandoned by the interviewee. These interviews were considered incomplete and were excluded from all aspects of the analysis. As an additional note, in one email interview the participant did not provide responses to several follow-up questions. This interview was also considered complete and included in all aspects of the analysis.

**Missing survey data.**   There were also several gaps in the screening and post-interview surveys. Participants were never required to answer any survey questions except to provide their ID number, so some questions were skipped or missed. Four participants completed an interview, but did not complete their post-interview survey (one audio, two email, and one survey). They were all compensated despite failing to complete the survey. Interviewers did not perfectly complete the task of filling out post-interview survey reports (see Technical Issues).

**Missing qualitative data.**   Despite implementing double-coding and double-transcribing, occasional errors were introduced when analyzing interview content. Most of these were discovered and addressed during the coding process. However, after the coding team completed

**Table 3. Summary of statistical tests.**

| Outcome Measure | All Conditions | Non-Survey Conditions | Details |
|---|:---:|:---:|---|
| Self-Disclosure | ✓ | x | SID survey instrument [32] |
| Honesty | ✓ | x | Self-reported survey response |
| Scheduling Rate | ✓ | x | Participants who scheduled interviews |
| Completion Rate | ✓ | x | Participants who completed interviews |
| Interviewee Word Count | ✓ | ✓ | Word count of interviewee responses |
| Qualitative Codes | ✓ | ✓ | Count of all qualitative codes |
| Rare Codes (Std. Dev.) | ✓ | ✓ | Codes two standard deviations below the mean |
| Rare Codes (Quartiles) | ✓ | ✓ | Codes in lowest quartile of the distribution |

For outcome measures derived from the transcripts two tests were run. One across all conditions that excluded data from followup questions and another across non-survey conditions that included data from followups.

their work, they retired from the project team, and were not available to correct errors discovered much later. As a result, in a few interviews the occasional utterance was left without a corresponding code and in one case an entire audio transcript was not qualitatively coded.

## Quantitative analysis

We conducted an exploratory analysis to visualize our results and check for any confounding factors that might have introduced bias into the data. This included the influence of individual interviewers, participant demographics, participant familiarity with the technology platforms employed, and the number of logistical or technical issues that arose during the interview. No substantial differences were observed for these factors across mode.

Once these data checks were complete, we ran a total of twelve ANOVA tests across interview mode on eight outcome measures. A summary of these tests can be found in Table 3. Two ANOVA tests were run on several of the outcome measures. The first set of tests included all interview conditions, but excluded any followup questions since our survey conditions precluded followups. The second set of tests excluded the two survey conditions and included followup questions.

Since the unit of analysis in all ANOVA tests was at the interview level, any partial interviews were excluded (see Edge Cases). For all statistical tests, we report significant results found using $\alpha = 0.05$. With seven conditions, we adjust multi-way comparisons using Tukey's honestly significant difference test [44]. Our group sample sizes are slightly imbalanced throughout (see Recruitment) which reduced our statistical power.

## Validity and limitations

As previously stated, this study design was a process of balancing tensions in validity. We wanted to make the conditions controlled enough to allow for meaningful comparisons, but not so controlled that the assumed strengths of each interview mode were stifled. However, as a result, our study has many limitations from the perspectives of ecological and internal validity.

As mixed methods researchers, we acknowledge that there are drawbacks to performing statistical tests on qualitative codes. In particular, creating a codebook for qualitative coding is a process with low statistical resilience. Even if following our protocol and transcripts, a different group of researchers would likely arrive at a different codebook. This well-researched phenomenon means that our results offer an snapshot of a set of possible qualitative states that

may have been produced from our data [45, 46]. As such, the reproducibility of this study may be limited in this regard.

From a quantitative perspective, the relatively small sample size severely limits our statistical power. While we completed 154 interviews, the Audio (n = 18) and Video (n = 19) conditions in particular were lower than our sampling targets. However, we are also cognizant of the fact that in practice, many fields and methodologies often have sample sizes of 10–25 participants [47, 48]. As a result, if our work here shows few statistically meaningful differences in data equivalence, then studies in the wild are also unlikely to have differences as a practical matter.

Recruiting from Prolific Academic limits the generalizability of our results. For example, as our participants were digital workers who are usually paid to complete survey or similar text-based tasks, their propensity towards surveys may not align with other online research pools. In addition, our participants are likely especially susceptible to desirability effects. For example, when asked to self-report their honesty levels, we assured participants that their response would not affect their compensation or Prolific reputation. However, in open responses we saw occasional evidence that the uneven power dynamic was still salient to some participants. One Scheduled Survey participant noted, "I was honest because I really try to be honest now that I am in recovery. I was really excited to be invited to high paying interview and need the money so I don't want anything to get in the way of my ability to make money and contribute to my household." Although such factors are likely to play a role in other qualitative studies with monetary incentives, certain platforms and participant pools may not have the same level of effect.

## Findings

In the following six sections we detail our findings that cover a broad spectrum of factors to be considered when designing and conducting online interview studies. We report both our quantitative comparisons across mode and our qualitative findings based on the responses of the interviewees and interviewers who took part in the study. First, we find that participant demographics remained consistent across mode and demographic factors did not affect completion rates. However, we also observe that challenges pertaining to recruitment and logistics were higher in audio and video modes. Second, we discuss how rapport did not differ greatly among non-survey interviews, but anecdotally chat had a polarizing effect with some participants feeling like they could share more and others left feeling disconnected. Third, we observe that there is little evidence to support differences in self-reported honesty or social desirability bias across mode. Fourth, we show that empirically there were no differences in disclosure, self-reported or measured, across mode but at an individual level, participants reported being cognizant of mode and that it affected how much they were willing to share. Fifth, we find that participants did report statistically significant differences in perceived anonymity between the non-anonymous and anonymous chat conditions. However, between the anonymous conditions we do not observe differences in perceived anonymity despite differences in real threats of de-identification (e.g. revealing one's face in a video interview). Sixth, we find that while structurally we observe statistically significant differences in word count across mode, thematically we do not observe differences in data-equivalence upon applying qualitative codes.

### Demographics, recruitment, and logistics

We screened 1,240 people and invited 310 eligible participants to an interview. Of this group, 200 initiated the interview scheduling process and 154 completed interviews. Overall, we observe that mode does not affect the ability of different demographic groups to schedule and

complete the interviews. However, we do find that the overall recruitment rate, completion rate, cost, and technical difficulties encountered vary across mode.

**Participant demographics.** The screening pool of 1,240 participants spanned a wide age range with the youngest participants falling in the 18–24 category and the oldest in the 65–74 category. Overall our sample skewed younger, with over one-third of the interviewees in the 25–34 age range. Around 60% of our sample had never been married, likely a result of our skew in age. Of those screened, men were slightly over-represented, with a breakdown of 51% men, 46% women, 1% non-binary, genderfluid, or genderqueer (*non-binary+*) and 1% unknown. However, this trend reversed for completed interviews where 52% of our sample were women and 45% were men. Table 4 summarizes the demographic breakdown of

**Table 4. Participant demographics.**

| (**a**) Gender | | | |
|---|---|---|---|
| *Non-binary+* includes non-binary, genderqueer, and genderfluid identities. | | | |
| Gender | Screened | Invited | Interviewed |
| Man | 640 (52%) | 153 (49%) | 69 (45%) |
| Woman | 570 (46%) | 152 (49%) | 81 (53%) |
| Non-binary+ | 20 (2%) | 4 (1%) | 3 (2%) |
| Unknown | 10 (<1%) | 1 (<1%) | 1 (<1%) |
| Total | 1240 (100%) | 310 (100%) | 154 (100%) |

| (**b**) Age | | | |
|---|---|---|---|
| *Unknown* indicates non-response, were excluded from the study. | | | |
| Age | Screened | Invited | Interviewed |
| 18–24 | 330 (27%) | 65 (21%) | 32 (21%) |
| 25–34 | 508 (41%) | 126 (41%) | 56 (36%) |
| 35–44 | 226 (18%) | 67 (22%) | 35 (23%) |
| 45–54 | 89 (7%) | 30 (10%) | 20 (13%) |
| 55–64 | 62 (5%) | 19 (6%) | 9 (6%) |
| 65–74 | 21 (2%) | 3 (<1%) | 2 (1%) |
| Unknown | 4 (<1%) | 0 (0%) | 0 (0%) |
| Total | 1240 (100%) | 310 (100%) | 154 (100%) |

| (**c**) Race and ethnicity | | | |
|---|---|---|---|
| *Options are not mutually exclusive, except *White alone* and *Unknown*. | | | |
| Race | Screened | Invited | Interviewed |
| White | 848 (68%) | 222 (72%) | 115 (75%) |
| White alone* | 839 (68%) | 220 (71%) | 115 (75%) |
| Latino | 120 (10%) | 25 (8%) | 11 (7%) |
| Black or African American | 104 (8%) | 28 (9%) | 12 (8%) |
| Asian | 84 (7%) | 15 (5%) | 6 (4%) |
| Native Hawaiian, Pacific Islander | 40 (3%) | 12 (4%) | 2 (1%) |
| Middle Eastern, North African | 39 (3%) | 13 (4%) | 4 (3%) |
| Unspecific multiracial | 29 (2%) | 7 (2%) | 3 (2%) |
| American Indian, Alaska Native | 24 (2%) | 7 (2%) | 2 (1%) |
| Unknown* | 18 (1%) | 4 (1%) | 1 (<1%) |
| Some other race | 6 (<1%) | 2 (<1%) | 1 (<1%) |
| Total | 1240 (100%) | 310 (100%) | 154 (100%) |

Demographic counts of participants who were screened, who were eligible and invited, and who completed an interview. Percentages are column-wise. For example, of those who were screened, 10% were Latino. Of those who were invited to interview, 7% were Latino.

the screening pool, those who were invited to an interview, and those who completed their interview.

In the screening survey, participants were presented with two freeform text boxes to allow participants to self-report race, ethnicity, and any cultural group they felt was significant to them. Answers ranged from "Mostly midwestern caucasian american, with some asian american and malaysian aspects thrown in" to "Black, cosmic" to "White, Burning Man." Full responses were shared with interviewers to provide context about their interviewees. To provide a summary and assess the distribution of our sample, we adapted the US Census Bureau guidelines to standardize these responses [49]. Our estimates of race and ethnicity roughly track the 2010 census data. However, our sample over-represents White participants and under-represents black, African American, and Latino participants.

Our exploratory data analysis indicated that, after random assignment, the gender and race of participants were distributed relatively evenly across modes. However, on average, interviews conducted with black participants were shorter in terms of word count ($\mu = 585$ $\sigma = 347$) and yielded slightly fewer qualitative codes ($\mu = 14.8$ $\sigma = 4.2$) than Latino (word count: $\mu = 974$ $\sigma = 754$, codes: $\mu = 17.4$ $\sigma = 6.1$) or white (word count: $\mu = 870$ $\sigma = 762$, codes: $\mu = 18.1$ $\sigma = 7.4$) participants, but were generally within a standard deviation of each other. Since the number of black and Latino participants were relatively small (1–2 participants of each minority group per condition) we were unable to do any meaningful sub-group analysis across mode. As a result, we cannot say with any certainty whether mode would be more or less of an effect for these groups. This raises some question of external validity that indicate our findings should be generalized cautiously, particularly for studies focusing on minority communities (see Accessibility for additional discussion of external validity).

Age was less evenly distributed which raised concern of internal validity. For example, we examined whether age might affect completion rates in different interview modes. We hypothesized that older participants would be more likely to complete email or audio interviews where younger participants would be more likely to complete chat interviews. However, we did not observe an effect and, in general, we found no evidence to suggest that demographics had an impact on scheduling rates, completion rates, or completion by mode.

**Technology and environment access.** Access to the necessary technology and environment to participate in an online interview is a limiting factor when recruiting participants and is a more restrictive factor for audio and video interviews. Of our screening pool, 99% reported having access to a computer with a keyboard, as opposed to a touch screen device like a smartphone or tablet. A keyboard was required for eligibility in this study because it is very useful for text-based interviews, though not a prerequisite. However, only 70% of those screened had access to a webcam. This precluded participation in video interviews conducted using a computer, however these participants might have a tablet or smartphone with a camera suitable for video interviews.

In addition to technology, Jenner and Myers (2018) noted that a private space was a more salient factor than mode for their participants [7]. Of our respondents, 80% had access to a quiet, private room. More than 90% reported that they usually completed Prolific tasks at home, followed by 8% at an office space, and a final 2% mix of specific locations such as coffee shops. However, in interviews, several participants were concerned about being overheard in their space. In one case, an Audio participant was concerned about responding to a question about sex. While laughing, they responded "oh my god. can I answer this?. . .I'm not alone in the house." The Audio participant eventually opted instead to write their response in the chat box. In another case, a Video participant got up to close the door, saying "I have a 16-year-old son who might walk by, so. . . [laughs]". In follow-up surveys, nine participants (6% of 125)

reported that their conversation was easily overheard and six (4%) reported that they were interrupted.

While we did not collect socioeconomic data directly, the high level of access to technology and private space, combined with a sample that is disproportionately white, indicates that our sample likely overrepresents individuals with a higher socioeconomic background. Since participants' race was distributed relatively equally across mode and access to suitable technology and private space was a prerequisite for eligibility, we conclude that this skew does not affect the internal validity of the study or bias our findings related to interview mode. However, this does challenge the external validity of our results. While our study draws on participants from a large online panel of respondents, it is likely that there are other populations for which particular modes may have more of an effect. This is indicative of a broader issue of representation in online studies that researchers should take into consideration when deciding to conduct their research online. Since participants in most online panels are digital laborers, it's likely their access to technology does not represent the access of the broader population, especially those of lower income and members of marginalized communities. At the same time, other online research studies using large panels of digital workers are likely to encounter similar bias. Therefore, researchers using similar recruitment methods can likely apply our findings directly. However, those working with participants of lower income, those with less access to online resources, or members of marginalized groups should use our results cautiously.

**Willingness to participate differs by mode.** In the screening survey, participants were asked: "What types of online interview are you willing to complete? Select all platforms that apply." From the participant's responses we found large differences in willingness to participate across modes. As displayed in Fig 1A, participants were less willing to complete audio and video interviews compared to survey, email, or instant chat. While access to the requisite technology and to a private environment factor into this, those factors alone do not explain the strong preferences of participants. Of respondents with webcam and computer access, only 49% were willing to do a video interview. It is likely that the additional logistical burden and lower level of anonymity also contribute to the trend. In addition, participants might also have been less willing to participate in audio or video interviews because the platforms are not conducive to multi-tasking and require a participant's full attention. Overall, unwillingness to participate in a video was the largest disqualifying factor in our pool.

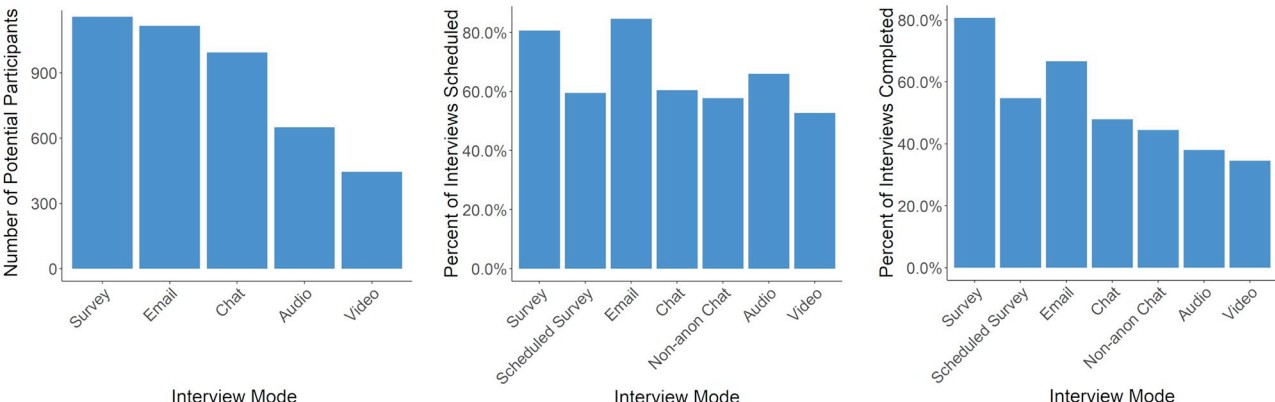

**Fig 1. Differences in recruitment and logistics across mode.** (A) Differences in willingness to participate across interview mode ($n = 1240$). (B) Differences in the scheduling rate, comparing those who scheduled an interview ($n = 200$) to those who were invited ($n = 310$), across mode. (C) Differences in completion rate, comparing those who scheduled an interview ($n = 154$) to those who were invited ($n = 310$), across mode.

Respondents were also asked about their willingness to be recorded as a part of the interview. Since "in most qualitative traditions it is strongly preferable, if not essential, to have a full record of each interview," this is an important factor for qualitative research teams to consider when recruiting [1, p. 44]. Specifically participants were asked "are you willing to conduct an interview where the video is recorded?" Only 45% responded affirmatively. However, this sentiment appears to be driven largely by the video mode itself rather than the recording. Of those who were willing to do a video interview, 95% were willing to be recorded. However, it is possible that respondents assumed that all interviews would be recorded when indicating their willingness to participate in interview modes; we did not specifically ask about the counterfactual of participating in a non-recorded video interview.

**Scheduling rates differ by mode.** Although all participants who were eligible to participate in the study indicated they were willing to participate in any type of interview, we observed substantial differences in the rate that participant's scheduled interviews across mode. As shown in Fig 1B, 310 participants received an email inviting them to schedule an interview. Of those, 200 (65%) initiated the interview process. We found a statistically significant difference ($F_{6,303} = 2.657$, $p < .05$) in the scheduling rate across mode. The post-hoc Tukey pairwise comparisons showed that the video and email conditions differed significantly at $p < 0.05$, with a 20% drop in engagement from email to video. The other 20 comparison combinations were not significantly different. A summary of the results can be found in S1 Table.

Note that the participants received their interview invitation via email, thus the increased scheduling rate for the email condition may be in part the product of platform familiarity or self-selection (i.e., people that checked their email are more likely to use email). While not statistically significant, the drop in engagement from the Survey condition to the Scheduled Survey condition illustrates the cost of scheduling. With all other factors held constant, asking a participant to schedule a time to take their survey, as opposed to taking it immediately, appears to have caused a drop in engagement.

**Completion rates differ by mode.** Similar to scheduling rates, we also observed statistically significant differences in the rate that participant's completed interviews across mode. Overall, the completion rates aligned with the overall screening pool's willingness to complete an interview by mode. As shown in Fig 1C, participants completed surveys and email interviews at higher rates than the chat conditions. Audio and video interviews were completed at the lowest rate. An ANOVA test ($F_{6,303} = 4.46$, $p < .001$) and subsequent Tukey revealed statistically significant differences in survey versus audio, video, chat, and non-anonymous chat, as well as in email versus video. There was only one instance where an interviewer missed a scheduled interview. However, the single occurrence indicates that interviewer error was not a substantial factor in completion rates.

The chat and non-anonymous chat conditions had nearly identical scheduling and completion rates. The invitation emails and scheduling site did have small differences between these conditions to emphasize the lack of anonymity in the non-anonymous condition. This suggests that emphasizing anonymity had little effect on recruiting for this population.

**Estimated cost.** While there are many considerations that a researcher must weigh when designing a study, cost is often one of the most important driving factors. Based on our experience running this study, we found that the cost of conducting an interview study varies widely depending on mode. These estimates, summarized in Table 5, are based on conducting 25 interviews using our interview script. To provide context, the median duration of a video interview, the mode that most closely mimics an in-person interview, was just over 17 minutes. In our our study, we compensated interviewers and participants $10 equally across mode, regardless of the duration of the interview. However, in designing an interview study researchers are likely to set compensation according the the expected duration of an interview.

**Table 5. Estimated costs to complete interviews by mode.**

| | | Survey | Email | Chat | Audio | Video |
|---|---|---|---|---|---|---|
| Parameters | Screening Eligibility Rate | 94.7% | 91.1% | 81.4% | 53.1% | 36.4% |
| | Completion Rate | 80.6% | 66.7% | 47.9% | 38.0% | 34.5% |
| | No-Show Rate | 0.0% | 15.4% | 26.1% | 57.9% | 47.4% |
| | Interview Duration (Minutes) | 10.0* | 14.5† | 46.2* | 16.7* | 17.2* |
| Costs | Screening Compensation | $33 | $42 | $65 | $124 | $200 |
| | Interview Compensation | $50 | $145 | $462 | $167 | $172 |
| | No Shows‡ | $0 | $12 | $65 | $50 | $41 |
| | Transcription & Processing | $60 | $60 | $60 | $225 | $225 |
| | **Total** | **$143** | **$259** | **$652** | **$566** | **$638** |

Estimated costs across mode based on the screening rate, now show rate, completion rate, and interview duration of each interview type. Costs were computed using a target of 25 completed interviews, $1 compensation for screening, and an equivalent $12 per hour compensation for both the interviewer and interviewee.

*Measured using median interview duration to account for several outliers.

†Estimated using a timed mock interview that reproduced a median length email transcript.

‡Assumes the interviewer is compensated for lost time due to a no show.

Therefore, in this estimate we assume participants are compensated $1 for completing the screening survey and both interviewers and interviewees are compensated at a $12 per hour rate upon completion of an interview. These estimates were calculated using five factors that we found varied widely across mode. The first was the screening eligibility rate: the percent of eligible participants (18+ with access to a computer) that are willing to participate in a given type of interview. This affects the number of participants that must be screened and compensated for completing screening. The second was the completion rate: the percent of eligible participants that go on to complete an interview. This also affects the number of participants that must be screened. The third factor was the no-show rate: The proportion of participants who do not show up at their scheduled interview to those that do. This incurs a cost assuming that the interviewer is compensated for the lost time that they had blocked off for the interview even though it did not occur. For some studies this might not be applicable. The fourth variable was the duration of the interview which influences the compensation paid both the interviewer and interviewee. With the exception of email interviews, we used the median interview duration for scheduled, anonymous interview conditions. Email interviews, which are inherently asynchronous, could not be measured in the same fashion. For this mode, the research team timed a mock interview that reproduced an interview transcript of median length. The fifth and final factor was the time required to process transcribed data, in the case of audio and video modes, and to process all of the transcripts (e.g. compile, clean, format, and anonymize).

We should note that our estimates do not account for the cost of recruiting participants to take the screening survey or fees for using a participant pool or crowd working service. In addition, they do not include other potential costs such as transcription, qualitative data coding, fees to use a survey platform, or the researchers' time.

With a total cost of $143, a survey is the least expensive way to collect data. Since there is no interviewer to compensate and, screening aside, no cost to a participant who does not show up or complete the survey this is not surprising. However, it is important to remember that this mode does not provide researchers with the ability to follow up immediately on a participant's response with follow-up questions (researchers would have to contact participants later to ask follow-up questions). On the other end of the spectrum, we find that chat-based interviews are the most expensive costing a total of $652. While chat interview do not require transcription,

the high cost is driven by a much longer interview duration compared to other modes. For long interviews, this may make chat a prohibitively expensive option. However, unlike other synchronous modes (audio and video) which may have a shorter total duration, chat interviews afford the participant and the interviewer with the flexibility to multitask while completing the interview. This may allow for slightly lower hourly payments, but more importantly it may help with scheduling constraints and contribute to a higher completion rate and lower no-show rate than in audio and video.

Since audio and view interviews incur additional costs related to transcription, we found that the total cost of audio and video interview, at $566 and $638 respectively, to be comparable to chat interviews. While our study used an automated transcription service included in our institution's Zoom subscription, our research assistants spent an additional 25 minutes verifying and correcting each automated audio and video transcript. This additional cost makes audio and video interviews almost as expensive as chat interviews despite being almost three times shorter to complete. Between audio and video, we find that audio interview are slightly less expensive. The difference is driven primarily by the difficulty in recruiting participants who are willing to be in a recorded video interview. As a result, researchers must weigh the added logistical burden of video interviews with the possible benefits of a richer medium.

Email, an infrequently used method for conducting interviews, provides an interesting option that researchers should give further consideration. In addition to its asynchronous nature we found that, surveys aside, email interviews had the highest eligibility rate, highest completion rate, and lowest no-show rate. As a result, email interviews provide a relatively cheap alternative to other interview modes at a total of $199. This is particularly advantageous option for researchers under substantial resource constraints.

**Effects of the Covid-19 pandemic.** The interviews in this study were conducted in the summer of 2019, before the COVID-19 pandemic began. The pandemic rapidly increased the adoption of online communication technologies, but also complicated the relationships users had with the platforms. Zoom, the platform used in this study, saw an enormous increase in users during the pandemic. However, the constant use of online communication platforms gave rise to "Zoom fatigue," a specific exhaustion caused by repeated video calls [50].

The implications of this phenomenon in relation to this study was not clear so in September 2020 we re-ran our screening survey with 100 new participants on Prolific Academic. We hypothesized that that the willingness to participate in modes like audio and video interview might have increased due to familiarity and access to the requisite technology and to a private space for online communication may have increased with people working from home. The new batch of results showed a slight increase in webcam access (+6%) but a decrease in access to private space (-3.9%). This indicates that working from home may have brought around changes in online communication technology, but the more people working in a shared living space during the pandemic decreased access to private space. Overall, we observed the willingness to participate in online studies dropped 3–7% depending on the mode. We performed eight two-sample *t*-tests comparing metrics between the survey results prior to the pandemic to those after. This included testing access to private space, webcam access, and willingness to participate in video, audio, chat, survey, email, and video interviews. None of the tests yielded statistically significant results. Full results tables are available in S16 Table.

**Technical issues were minimal.** After each interview and attempted interview, we surveyed the interviewer about the experience. Interviewers did not complete this protocol task perfectly and it was not required for either survey condition. As a result we collected 137 interviewer post-interview surveys, covering 129 unique participants, 106 of whom successfully completed their interview.

Of the 137 post-interview interviewer surveys we collected, only four reported that technical issues prevented an interview from proceeding. In two cases the participants used an anonymous email address provided by Prolific, but withdrew after reporting trouble with Prolific's email forwarding. In one case, a participant was unable to find the instructions or link for joining the video interview. In the final case, a chat interview was abandoned halfway through due to participant confusion with the platform.

There were five instances of dropped connections, but all five interviews were completed nonetheless. Audio quality issues were more frequent, with eight separate occurrences (21%) during the 37 attempted audio and video reports collected. In half of these cases interviewers reported that the interview was affected only "A little" and in the in the other half the interview was affected "A moderate amount." For example, after one video interview, interviewer 2 reported "moderate" issues with the audio quality, saying it "was not very clear at times so it was difficult to understand everything he said. He also had a hard time understanding my questions." In several of these instances, the transcription team was unable to transcribe parts of the interviewee's response as a result of the quality of the recording. However, in none of these cases did the interviewer report that the interview was affected "A lot" or "A great deal". All of the affected interviews were completed in their entirety.

Of those who completed their interview and post-interview survey, interviewees who used the audio, video, and chat platform reported less familiarity with the technology used. Fig 2 shows the distribution of interviewee responses when asked how familiar they felt with the online platform used in their interview.

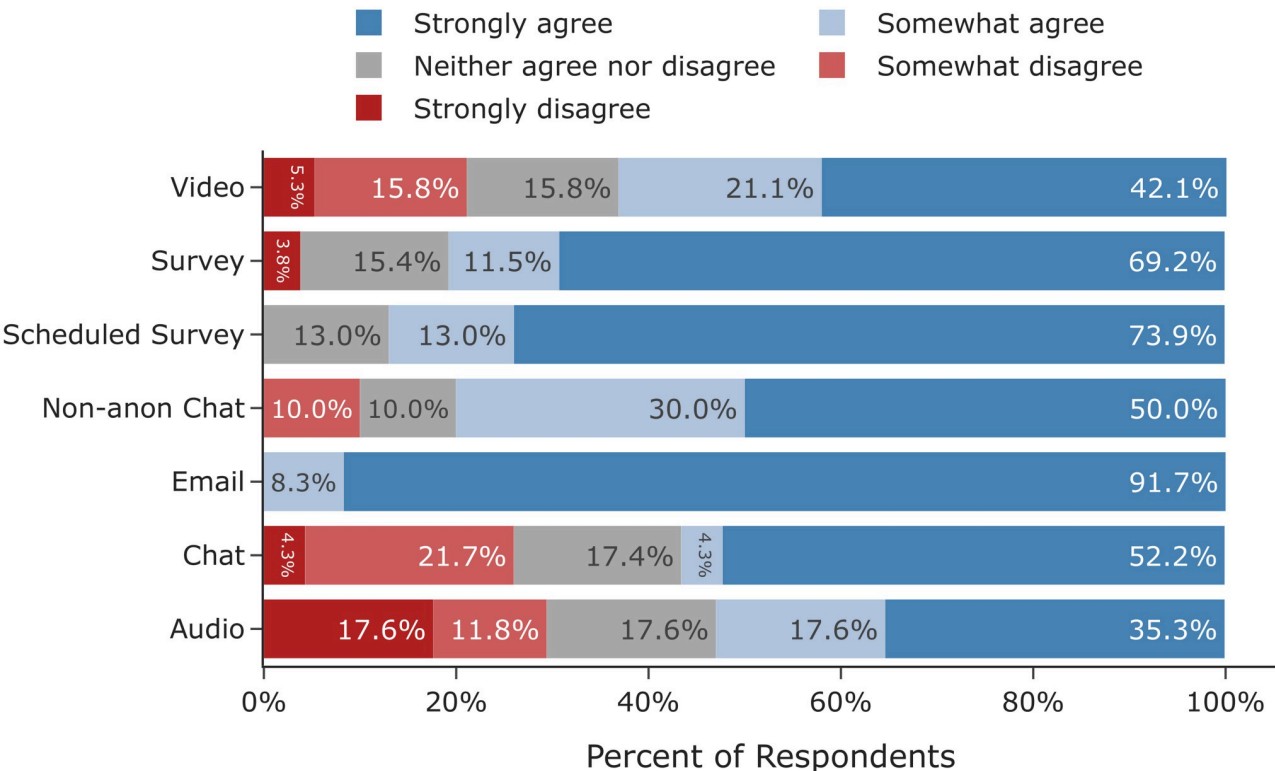

**Fig 2. Interviewee's familiarity with online interview platform.** Interviewees rated their level of familiarity with their mode's platform and were asked to agree or disagree with the following statement: "I felt familiar with [platform], used for the interview." While users were less familiar with the audio, video, and chat platform, participants did not report differences in ease of use across modes.

Despite the relative lack of familiarity with the video, audio, and chat platform (Zoom), in general participants overwhelmingly thought the platforms were easy to use across modes. This indicates that neither technical problems nor usability of online communication tools were a substantial barrier in our study. However, our sample may be more familiar with online platforms than the general population.

## Rapport

Excluding the survey conditions in which there was no interviewer, we saw no evidence to suggest that mode impacted interviewee discomfort or rapport in a systematic way. However, individual participants offered varying opinions about rapport and mode in their open-ended responses.

**Interviews had reasonable rapport.** Across the non-survey conditions, the mean rapport, based on a 4-question subset of the Arizona Clinical Interview Rating Scale, was $\mu = 15.8$ out of 20 ($\sigma = 2.7$) [12]. We examined the rating and individual items by mode, but observed no systematic differences for those 103 interviews. When we included the survey conditions, unsurprisingly, there were large differences in the rapport metrics. For example, a majority of survey participants reported that the survey was unsupportive.

**Interviewees felt ill at ease regardless of mode.** In the post-interview surveys, both the interviewer and the interviewee were asked two questions from Hill and Hall (1963) [8]: "How often did you feel ill at ease during the interview?" and "How often did the [interviewer/respondent] seem ill at ease?" As might be expected from an interview on sensitive topics, many interviewees reported having felt ill at ease at least "a number of times" during their interview. However, very few felt uncomfortable during "almost all" of it. In breaking this down by mode, as shown in Fig 3, we observed no clear patterns in the distribution of responses.

Despite feeling ill at ease, most interviewees still indicated that the interview was a positive experience. One audio participant summed up the interview saying, "It was interesting, slightly uncomfortable, but still fun."

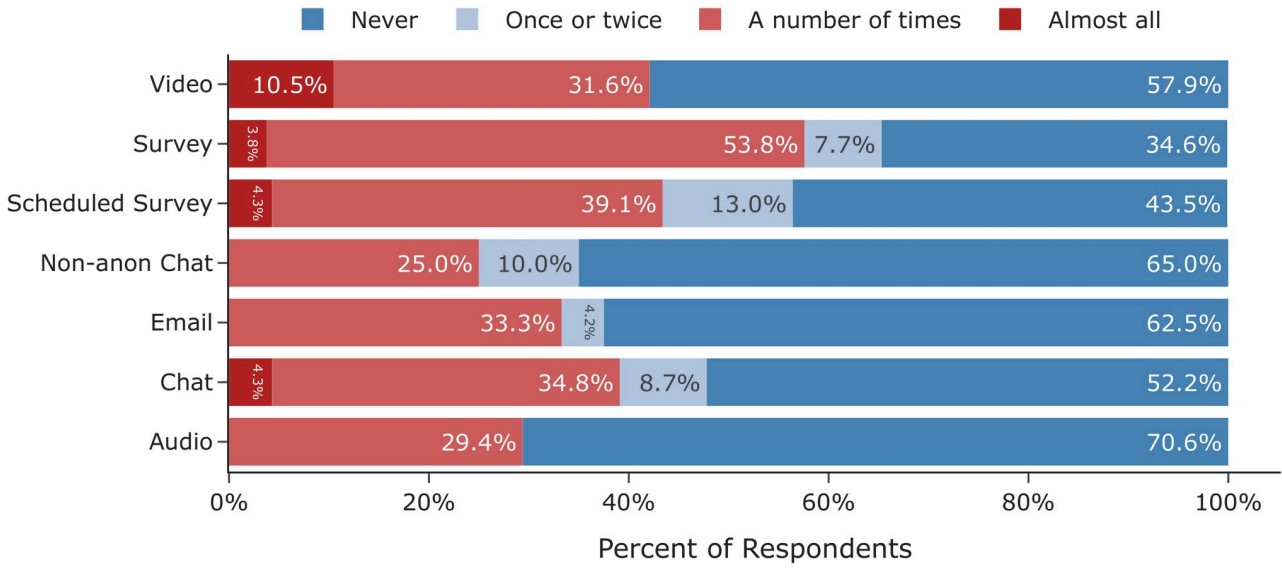

**Fig 3. Interviewee's self-report of feeling ill at ease.** In the post-interview survey interviewees were asked to respond to: "How often did you feel ill at ease during your interview?" In general, interviewees felt uncomfortable at least some of the time which is understandable given the sensitive nature of the questions asked.

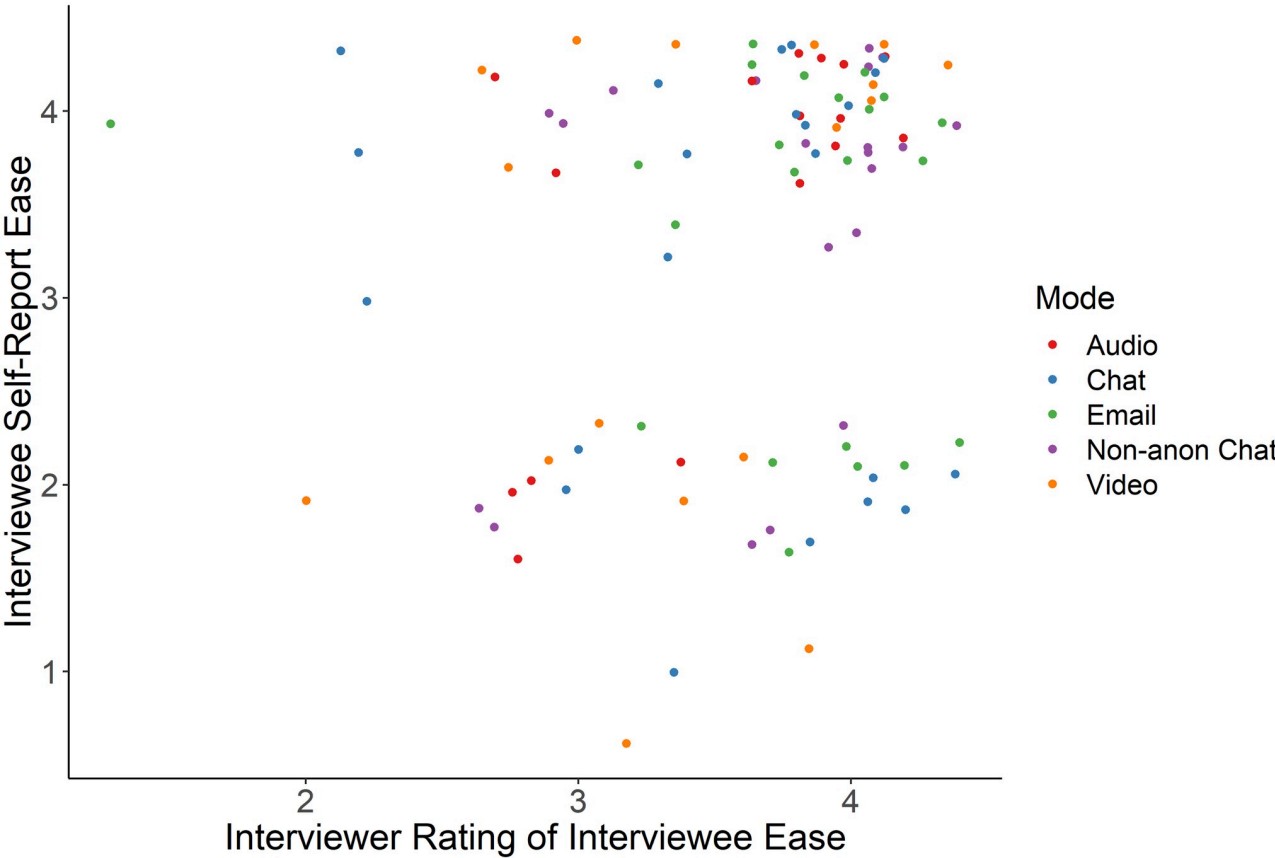

**Fig 4. Comparison of perceived and actual interviewee comfort.** Comparing interviewer (x) and interviewee ratings (y) of how ill at ease the interviewee appeared or felt reveals some overestimation by the interviewer. However, no patterns related to mode or interviewer emerge. There is a small amount of noise added to the points to allow for a visible scatter. Points are color-coded by interview mode.

**Interviewers may overestimate Interviewee comfort.**    In asking both interviewers and interviewees about their rates of feeling ill at ease, we were interested in whether interviewers were able to reasonably pick up on interviewee discomfort. Anecdotally, Interviewer 3 noted in one post-interview survey that they had difficulty establishing rapport in a chat interview: "They were very neutral in their answers, so it was hard to discern their emotions over chat." Fig 4 shows how the interviewer rated the interviewee's comfort level compared to how the interviewee rated themselves. In the jitter plot, the survey responses were converted to a number between one and four (one: *Never*, four: *Almost all*). Overall, interviewers were more likely to overestimate than underestimate interviewee comfort.

To examine whether the mismatch in perception was more or less likely by mode, we calculated the distance between ratings, and plotted them by mode. We excluded surveys in this case because the interviewer had no insight into the interviewee's experience. As the Fig 4 shows, we observed no differences across mode. In addition, the overestimation had no ties to specific interviewers and was observed equally among them.

**Chat modes could help or hurt rapport.**    In post-interview surveys, interviewees who commented on mode and rapport often did so for chat conditions. This could be because of novelty effects (as chat isn't a usual mode for interviews) or because interviewees had strong feelings about the mode. Anecdotally, chat seemed to be polarizing. Some interviewees felt it was easier to share, while others felt disconnected. One interviewee in a chat condition said, "It

felt like I was talking to an AI," while another in a non-anonymous chat said "The chat format worked well. I was surprisingly comfortable typing about these personal topics." Both interviewers and interviewees voluntarily observed that it was hard to gauge engagement in chat. One interviewee said, "The interviewer seemed a bit distracted or took long to respond at times." While in a check-in meeting, Interviewer 1 laughed that she "never had people with attitude on audio or video, but there was lots more on chats, because they have a screen to hide behind."

In a much more serious situation, Interviewer 2 felt very uncomfortable in one chat interview when a participant expressed suicidal thoughts. The interviewer recalled that the participant "said he hates living life. . .being a chat, I wasn't really sure how to respond to that. I think I just said, 'thanks for sharing.'" In cases like this where emotionally distressing issues arise, we had explicitly instructed interviewers that while they could offer assistance, they should thank the participant for their honesty and "gently disengage" from the topic since they did not have professional training as a therapist. While the interviewer followed those instructions, the lack of rapport and other audio and visual cues in the text-based interview likely affected the interaction and led to the interviewer disengaging less gently than in an optimal situation. As per the study protocol, the interviewer reported this issue to the research team and it was escalated to the principle investigator. The interviewer met with a member of the research team and discussed the interview, how they felt afterwards, and if they needed any additional support. The interviewer reported that they were emotionally okay and were fine to continue conducting interviews.

**Motivation not affected by mode.**   To gauge participant motivation, we asked interviewers to rate how motivated the interviewee appeared during the interview and asked interviewees to self-report. This was done using a 5-point Likert scale based on similar measures in Hill and Hall (1636) [8]. Overall, both the interviewees ($\mu = 1.79$, $\sigma = 0.77$) and interviewers ($\mu = 2.04$, $\sigma = 1.08$) felt that the interviewee were relatively motivated during the interview. On average, this fell roughly between "very motivated" and "motivated" or "motivated" and "neither motivated nor unmotivated" respectively. In examining the level of motivation across mode, we did not observe any substantial differences.

## Honesty

In this section we report the participant's self-reported honesty from 3 questions that were included in the post-interview survey. As a corollary, we measured the level of social desirability bias using the Balanced Inventory of Desirable Responding (BIDR-16), a 16-question survey instrument [38].

**Honesty does not differ by mode.**   In the post-interview survey we were interested in honesty as it relates to openness and truth-to-self, rather than honesty related to some objective truth. Interviewees self-reported withholding information in 16% of interviews (25 of 152) and altering the details of their responses 7% (11 of 152). However, it is not entirely clear if the latter was a result of interviewees wanting to hide information or a result of our encouragement for interviewees to protect the identity of themselves or others by using pseudonyms. In addition to these two questions, interviewees were asked to respond on a 5-point Likert scale to the statement "During the interview, I was not as honest as I could have been." As shown in Fig 5, interviewees generally disagreed with the statement and rated themselves as honest.

After converting the Likert to a numerical scale, running an ANOVA test ($F_{6,145} = 0.25$, $p > .5$) showed no significant differences in self-reported honesty across mode.

**Little evidence of social desirability bias.**   While not one of our outcome measures, we wanted to ensure that social desirability bias was not substantially different across modes such

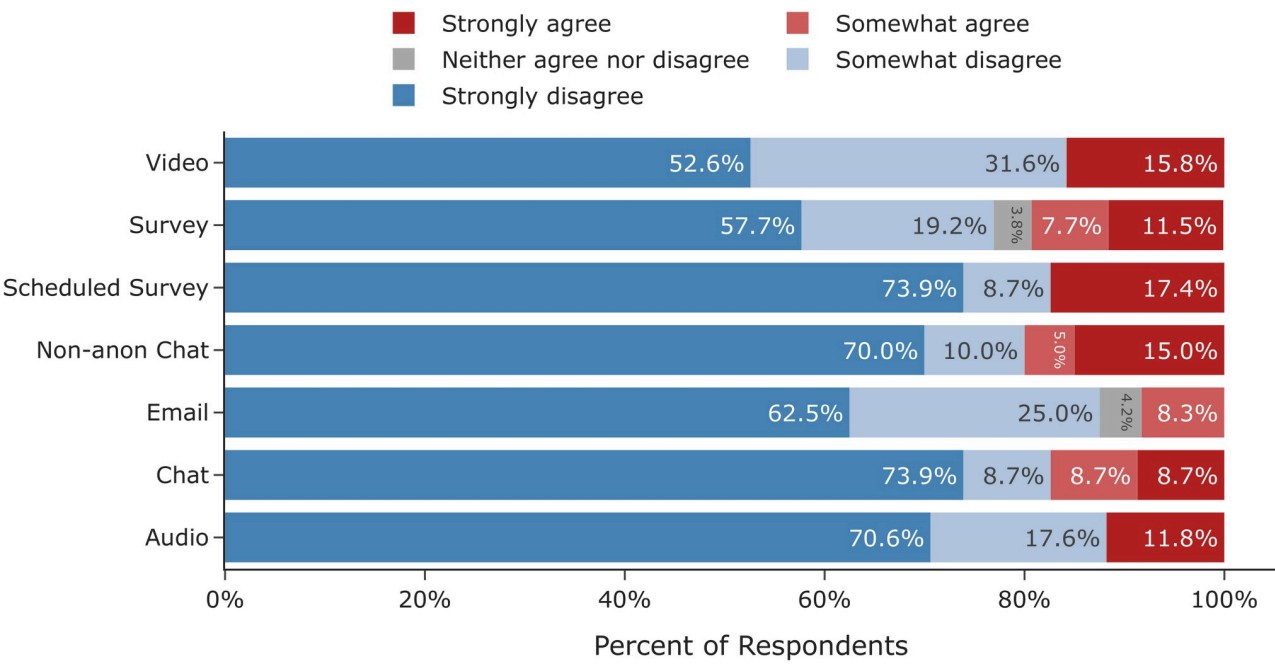

**Fig 5. Interviewee's self-reported honesty.** Interviewee's were asked to rate themselves based on the following statement: "During the interview, I was not as honest as I could have been." No statistically significant differences were found across mode.

that it might affect our other outcome variables. Using the BIDR-16 scale, we found an average score of 3.7 ($\sigma = 0.8$) on a 7-point scale. The distributions of BIDR-16 scores were incredibly consistent across mode. This suggests that none of the modes uniquely invited socially desirable responding bias.

## Self-disclosure

In previous work, researchers have evaluated self-disclosure and intimacy with a variety of measures. We used a few different approaches to operationalize this concept and examine it by mode.

- *SID index*: In the post-interview survey, interviewees responded to a self-disclosure index [20].

- *Depth*: Based on previous work, we considered cases of limited and exceptional self-disclosure [7].

- *Breadth*: We used the number of qualitative code categories as an indicator of self-disclosure breadth.

**Self-disclosure index does not differ by mode.** Our first assessment of participant self-disclosure used the Sensitive Information Disclosure (SID) scale that was included as a part of the interviewee followup survey. In their development of the 12-question scale, Pickard et al (2018) noted that the instrument "can be used to derive insights previously masked in similar studies comparing disclosures of sensitive information across interview modes" [20]. Within our sample we observed a normal distribution of scores with a mean of 2.81 ($\sigma = 0.48$) out of 5, where 1 represented a low level and 5 a high level of self-disclosure. However across modes, we

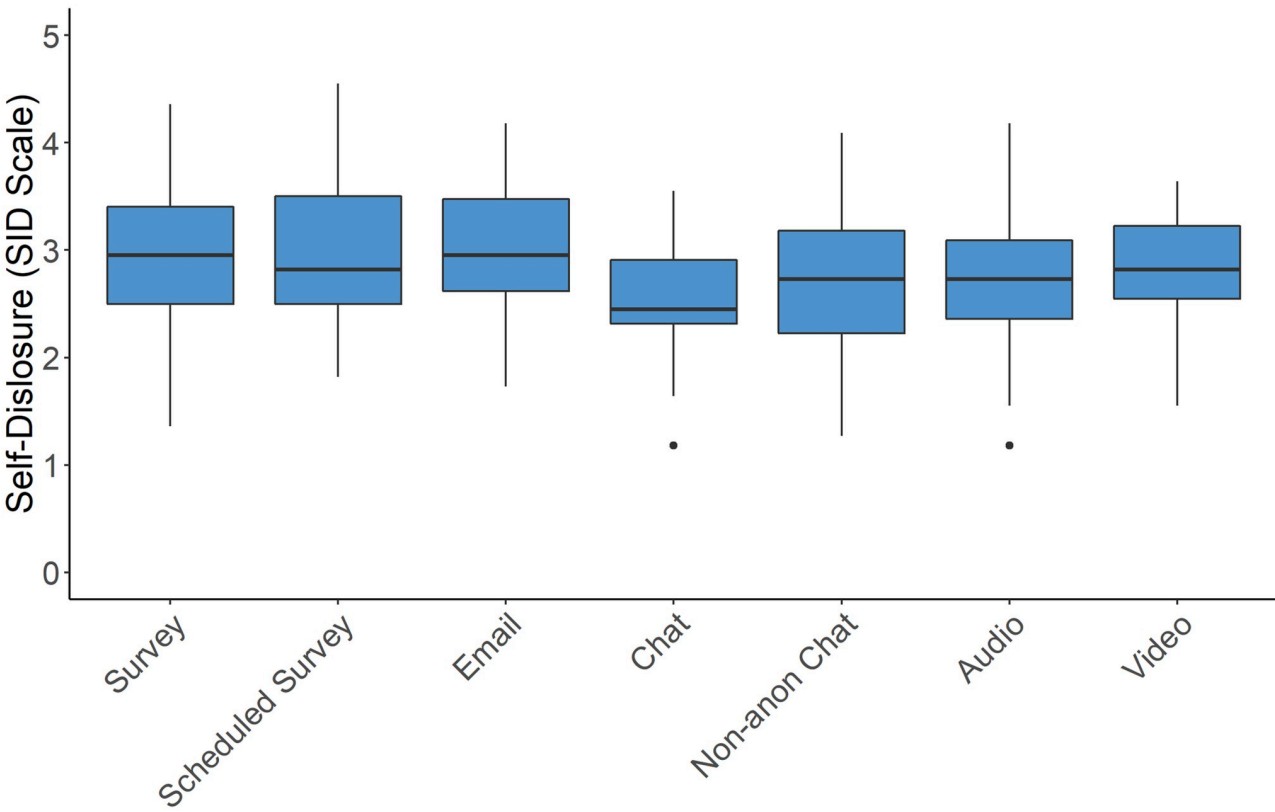

**Fig 6. Interview self-reported self-disclosure.** Interviewee scores on the Sensitive Information Disclosure (SID) index by mode. We observe no meaningful differences across mode.

did not observe any statistically significant differences ($F_{6,145} = 1.79$, $p > .1$). The distribution of scores by mode is illustrated in Fig 6.

**Disclosure depth was difficult to define.** The second self-disclosure measure we evaluated was disclosure depth. However, we encountered difficulties in defining an appropriate method to apply this metric to our results. While others have measured self-disclosure by asking third-parties to rate disclosure sensitivity or depth, the scales we encountered were not sensitive enough to detect differences for our interview with inherently sensitive topics [51]. We initially hoped to define our own scale for disclosure depth, and piloted several attempts, but were never satisfied with the reliability or interpretability of the emerging scales. For example, co-author Bernagozzi, a member of the coding team, wrote a coding reflection that discussed the difficulty of relying on hesitation as one indicator of limited disclosure, an idea repeatedly proposed:

> sometimes, the [interviewee] changes their mind or switches gears in the middle of what they are saying. Often, I attribute this to them "thinking out loud" as they are answering the questions, but. . .it could also indicate that they are holding something back. . .Other folks will decide mid-answer that another response would fit the question better and they will switch tracks, but I don't necessarily think they are being dishonest.

Therefore, we looked to Jenner and Myers (2018) [7], who analyzed their video and in-person interview transcripts across mode with special attention to identifying instances of

exceptional or limited disclosures. Members of the interviewer, coding, and research teams all hypothesized that the question related to sex had the greatest variability in disclosure. For example, in an early post-interview survey, Interviewer 3 volunteered the observation that in her interviews so far, ". . .it seems like the sexual fantasy question is something they tend to pass over with their answers" or "they responded quickly, honestly, and with detailed answers." One survey participant noted in their final open response, "I really don't want to talk about sexual fantasies with anyone I'm not in a relationship with. Way, way too personal, even anonymously."

Similarly to Meyers and Jenner, as the coding and research teams explored the sexual fantasy question each team independently found that it was more fruitful to identify cases of limited and exceptional disclosure rather than classify or rate general cases. In their codebook development, the coding team organically developed codes related to disclosure level. For example, the *elaborate* code identified responses with "an elaborate answer, including complete/detailed fantasy scene" while the *hesitation* code applied when the respondent "shows surprise or is unsure how to answer when hearing the question." After the second codebook draft, we asked the coding team to synthesize some of their codes related to limited disclosure, and they defined the code *soft pass* for when "the interviewee answers but does not directly or fully answer the research question." The research team took the final codebook and noted each code definition that involved limited (*soft pass*) or exceptional disclosure (*incest*, *infidelity*, *non-consent*, and *trauma*). These codes served as our markers of disclosure depth.

As expected, occurrences of limited and exceptional disclosure were very sparse in our corpus. In total, we found 29 exceptional disclosures and 10 limited disclosures. Given the low frequency, we were not comfortable doing a quantitative comparison of disclosure depth. We qualitatively examined the distribution and relevant transcripts, noting only that there was no evidence of limited disclosure in the survey conditions. We concluded that while it's possible that surveys prevent limited disclosure, it's more likely that indicators of limited disclosure, like non-verbal cues and pauses, do not present themselves in surveys. This conclusion is anecdotally supported by one survey respondent who, unprompted, decided to re-take the entire survey weeks later in order to disclose deeper detail (see Personal Information Sharing).

**Disclosure breadth did not differ by mode.** Previous work has operationalized disclosure breadth by counting the discrete, relevant topics introduced by the respondent [19]. We reasoned that our coding process organically identifies equivalent topics through the creation of code *categories*. For example, the categories for the codebook on death included *Emotions*, *Focus of emotions*, *Eras of life*, *Post-Loss actions*, *Spiritual views*, *Life experience*, and *Distance to death*. Treating code categories as our unit of measure, we found that the count and distribution of categories did not differ by mode.

**Individuals reported that mode impacted their disclosure.** Overall, the quantitative metrics we used to examine self-disclosure did not indicate any systematic differences across mode. However, at the end of interviews or in post-interview surveys, individual participants did sometimes volunteer their opinions on mode. Anecdotally, participants seemed more likely to volunteer a mode-related opinion after chat interviews (see Chat and Rapport). One non-anonymous chat participant wrote: "Given the format (text message), I tried to be as open as I could." In addition, after a non-anonymous chat interview, Interviewer 3 wrote:

> The interviewee told me that it was much easier for them to open up about these difficult topics over chat to a complete stranger than to talk to someone in person. I thought this was really interesting, and I didn't know how that level of anonymity affected what/how much we choose to share.

In contrast, one email participant explicitly described that they withheld or altered their response during the interview, but felt comfortable discussing them in a post-interview survey.

> I wasn't dishonest but I was super uncomfortable about the sexual fantasy question. I have a lot of insecurities about my body so I'm just not really open about sex because of that. And I don't like talking about it. I also wasn't as open about my depression as I should have been —I didn't mention the time I was laid off from work and didn't leave my apartment once in five months. Dunno why I'm now comfortable enough to admit this stuff. Maybe it's easier when I don't feel like I'm having a conversation with someone.

On one hand, we observe evidence that text-based communication may allow for greater disclosure, but at the same time the presence of an interviewer, even in as abstract a form as email, can have an affect on both the interviewee's level of comfort and their disclosure.

## Perceived anonymity

In post-interview surveys, interviewees responded to a 5-question perceived anonymity instrument developed by Hite et al. (2014) [32]. In general, the interviewees felt more anonymous than not, with an average score of 5.66 ($\sigma = 1.15$) on a 7-point scale.

We hypothesized that there would be differences between the chat and non-anonymous chat modes since the non-anonymous chat condition was designed to de-emphasize anonymity. Given the unbalanced sample sizes, $n = 20$ for Non-anon Chat and $n = 23$ for Chat, we tested this using Welch's t-test ($t = -2.0795$, $p < .05$). We found that perceived anonymity in the non-anonymous chat ($\mu = 5.44$, $\sigma = 1.15$) was statistically significantly lower than the regular (anonymous) chat condition ($\mu = 6.16$, $\sigma = 0.72$). Thus, in the chat conditions, when we removed advice on participating anonymously and addressed participants by name, it did decrease the perception of anonymity.

However, differences between other modes were lower than what we had anticipated. For example, we had anticipated that participants in the video condition might have more privacy concerns than participants in other conditions, as interviewers could see the interviewee's face even if they didn't know their name. Yet, we found no evidence to support this hypothesis and none of the video participants volunteered any information about privacy-related discomfort or concerns. This suggests that perceived anonymity was not dependent on "actual" threats of de-identification.

**Participants provided more personal information than requested.** All conditions except for the non-anonymous chat mode discouraged participants from sharing personally-identifying information and gave them instructions on how to prevent doing so (see Anonymity). However, there were cases where participants still provided us with personal information despite the instructions. In our scheduling sign-up form and interview platform, we auto-filled any personal information fields with reminders such as "Do not provide your name." Nonetheless, some participants chose to provide their name or use a personal email address rather than the anonymous one provided by Prolific. We logged 191 scheduling forms, including individual reschedules, noting whether the name fields appeared to be personally-identifying or likely pseudonymous. Of the 27 scheduling forms for the non-anonymous chat condition, 26 participants provided an identifying name and one provided a name that was possibly pseudonymous. In contrast, only 17 of 164 participants (10%) in the remaining (anonymous) modes chose to overwrite the default values with an identifying name. For audio and video modes, we checked whether the participants chose to provide an identifying name when prompted for a username by the Zoom platform. Of the 36 interviews, 30% shared a likely identifying name.

We do not fully understand why participants chose to provide personally-identifying information despite explicit instructions not to do so. It is possible that participants were confused by instructions, lacked tools to protect their identity if they desired, or felt ambivalent. One email participant did provide insights in a post-interview survey, reporting:

I thought the process for answering the survey through Prolific looked complicated, and I worried that I wouldn't get notifications for Prolific messages and would then not answer fast enough, so I didn't try that route and just set up my email address to be used. However, my email address contained identifying features, and I now kind of regret using it.

Another email participant responded:

I want to point out that I accidentally shared my first and last name with my interviewer when responding to her the first time, which is the reason for my answers to the privacy questions in this survey. That was my fault, but I don't believe that my personal identification will result in any privacy concerns.

**Individuals were conscious of anonymity.**    While the mode did not appear to have substantial effects on the perceived anonymity, interviewers and interviewees both indicated that anonymity was a factor during their interviews. Interviewer 2 noted in a post-interview survey that after answering all the questions, a non-anonymous chat participant "was concerned about how his identity would be kept confidential." In addition, a survey participant said, "I was honest in describing everything due to the fact that the survey is anonymous," and a scheduled survey participant said clearly, "I have nothing to hide when I am anonymous." Furthermore, after being asked the sensitive question pertaining to guilt, one video participant paused to confirm that the researchers were not collecting any personal information about them.

While researchers often collect identifying information from otherwise anonymous participants for logistical reasons, for example to facilitate scheduling and compensation, sometimes it is beneficial to shield participant identity even from researchers. For example, one scheduled survey participant said:

There were some questions, in which I could have gone more in-depth with, but because I know several people who go to [the university conducting the study] (and of which, one of whom is the subject of the questions of what I feel most guilty about), I felt as though I'd be discovered by them somehow, so I didn't feel comfortable in elaborating as much as I could have on some topics.

In one unusual case, a survey participant completed their interview and post-interview survey with quality answers and received their compensation. Curiously, five weeks later, the participant re-took both the survey and the post-interview survey. The participant was able to retake the surveys because our survey mechanisms were set up to minimize the use of trackers and thus did not automatically block retakes. The participant identified themselves by providing their anonymous Prolific ID in the survey, the primary identifier that was asked of all participants and was used to match between screening, interview, and post-interview tasks. In their second submission of the post-interview survey, the participant stated that they had purposefully decided to fill out the surveys a second time. This was the only instance of a participant completing an interview multiple times. The responses in both were similar except for a few instances related to the question about what the participant felt most guilty about. In the

first iteration, the participant provided a general response about the betrayal of trust and in the post-interview survey said, "I am not comfortable sharing the worst thing I've ever done in my life." However in the second survey, the participant disclosed a much more serious situation in which they expressed guilt over a past event involving sexual violence without disclosing any details. It seems the participant had been thinking about the interview response and, of their own volition and without notifying the research team, decided to re-take the survey interview in order to include this deeper disclosure. While it is difficult to interpret this story in the context of this study, it's interesting that the survey condition afforded the participant (1) a degree of anonymity and (2) the ability to take the full initiative in updating their response without needing to interact with anyone to do so. In the re-take the participant closed by saying, "I sure hope this is anonymous," illustrating that that the participant took a risk with disclosure despite having privacy concerns. As with other sensitive issues that arose during the course of this study, this case was immediately escalated to the principal investigator. The research team discussed this situation at length and considered what actions might be taken as a result. Ultimately, given the anonymity of the participant and the lack of any specific details surrounding the event, the team decided not to escalate further.

## Qualitative data equivalence

A central question in our analysis was not whether different interview modes produce different data, but whether those differences could in turn yield substantially different results. As such, we evaluated the data equivalence across modes using differences in structure (word count) and theme (qualitative codes).

Interviewer followup probes can have a substantial impact on the direction and length of an interview. Because surveys do not allow for followup questions, we reasoned that it would be difficult to interpret direct comparisons between surveys and other modes. We also reasoned that followup questions were the main vector for interviewer effects, as they were the one factor that interviewers had a large amount of control over. As a result, we decided to examine data equivalence factors on two different subsets of transcripts. First, we evaluated responses prior to any interviewer follow up questions across all seven modes. This subset allowed us to compare the "first impression" responses to questions, putting all the modes on a more controlled footing. Second, we excluded the two survey conditions and evaluated the participant's full response including followup questions. This subset limited the comparisons to the conditions that most resembled traditional interview modes.

**Word counts were higher for verbal modes.** Word count is a structural feature of interview data, but previous work has applied it to content-related constructs like self-disclosure depth or engagement [19]. Our own interviewers and coders also implied that they associated response length with content quality. For example, Interviewer 1 was concerned that in one chat interview "the answers were short and [the participant] not too willing." The coding team who worked on the question pertaining to sex even defined a code called *short* for responses that "give a very short answer." This further indicates that the team thought length had thematic implications worth investigating.

To calculate a word count for the main interview questions, we excluded non-interview related conversations at the start or close of interviews and included any non-verbal annotations that the transcribers noted. The mean across all completed interviews was 312 words ($\sigma = 318.4$) for the *followups removed* subset and 889 words ($\sigma = 629.0$) words for the *followups included* subset. In particular, the video and audio conditions had the most words, approaching twice as much as surveys and chats.

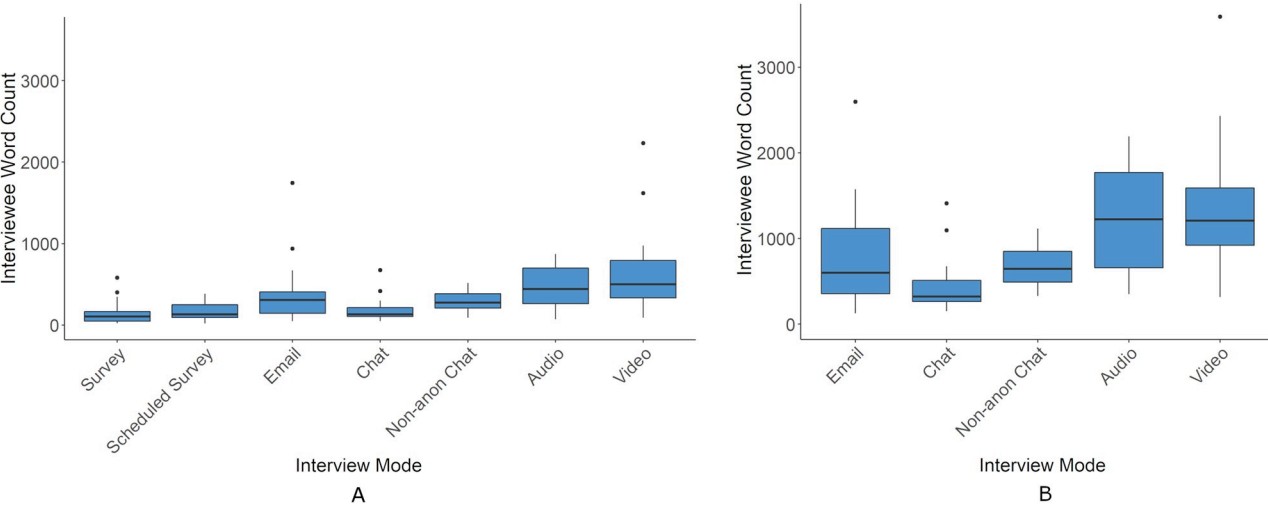

**Fig 7. Interviewee word count by mode.** (A) Interviewee word count, excluding responses to followup questions, by mode. (B) Interviewee word count, including followup questions, by mode. Excludes survey conditions where there were no followups.

We ran ANOVA tests on both subsets and found statistically significant differences across mode in both. Fig 7A shows the distribution of word count across mode for the *followups removed* subset. After the ANOVA indicated statistically significant differences ($F_{6,142}$ = 10.01, $p$ <.001), we evaluated the pairwise comparisons using a Tukey multiple comparison. This assessment showed that the audio and video conditions have a statistically significantly higher word count than chat, scheduled survey, and survey conditions at $p$ <.05 level. Video was also higher than non-anonymous chat and email ($p$ <.01). Detailed test results can be found in S7 Table.

Fig 7B shows the word count distribution for the *followups included* subset across mode. There were similar statistically significant differences found in the ANOVA test ($F_{4,94}$ = 10.61, $p$ <.001) and the Tukey multiple comparisons. We found video had a significantly higher word count than email, non-anonymous chat, and chat at the $p$ <.001 level. Audio also averaged significantly higher word counts than both chat ($p$ <.001) and non-anonymous chat ($p$ <.05). Detailed results are included in S7 Table.

Regardless of whether we included followup questions, audio and video participants had more to say than those in other modes. However, this may be a direct result of being able speak faster than one can type. Of the text-based modes, email had higher variability in word count. This suggests that the reduction of time-pressure inherent in the asynchronous email mode affords respondents the ability to write more than in other text-based modes.

**Code counts do not differ by mode.** In addition to word counts, we examined thematic differences across qualitative codes. According to the SAGE Handbook of Qualitative Data Analysis, the "generation of themes via coding. . . and categorization. . . is arguably the most common analytic approach taken by qualitative researchers using interviews" [41]. While not a representation of interpretive value, these codes serve as a proxy for the the breadth of content and themes covered during the course of an interview. The frequency and distribution of thematic codes, and differences in such across mode, provide richer insight into the quality of the data collected than word count alone. In this study, the coding was done by framing each qualitative code as one "meaning unit" of thematic analysis. We dis-aggregated these codes in order to compare unique code counts by mode, counting each code at most once in each

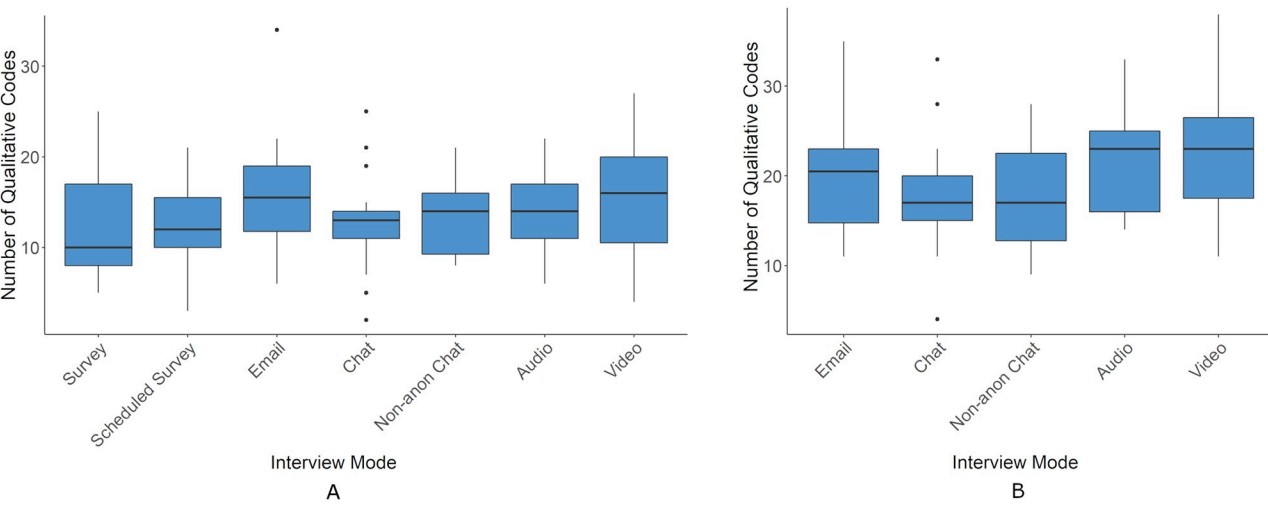

**Fig 8. Qualitative code count by mode.** (A) Qualitative code count, excluding codes derived from followup questions, by mode. (B) Qualitative code count, including codes derived from followup questions, by mode. Excludes survey conditions where there were no followups.

interviewee's response to a particular question. While simplistic, we use code count as a rough approximation of thematic analysis potential.

Fig 8A shows the code count distributions for the *followups removed* subset of transcripts across mode. Each condition averaged between 10 and 18 codes per interview. ANOVA tests revealed no statistically significant differences in code counts across mode ($F_{6,140}$ = 1.582, $p >.05$).

Fig 8B shows the code count distributions for the *followups included* subset of transcripts. With the followup question content included, code count averages increased to between 18 and 22 codes per interview. Just as in the other subset, there were no statistically significant differences across mode ($F_{4,94}$ = 2.369, $p >.05$).

**Code distributions differ by followup questions rather than mode.**　Stemming from this analysis, we conducted an additional exploratory qualitative analysis of the code distributions by question and mode. Fig 9 shows an example of a frequency table generated for the question on death. The figure highlights the relative proportion of interviews that had at least one code in a given category. We were curious whether certain codes were more or less likely to be present in certain modes. In the case of the death question, we found that survey participants did not detail their *life events* or volunteer a *focus of emotion* as often as other modes.

### Percentage of interviews (in each mode) that had at least one code in a given category

| Code category | Survey | Scheduled Surv | Chat | Non-anon Chat | Email | Video | Audio | Grand Total |
|---|---|---|---|---|---|---|---|---|
| *distance* | 56% | 70% | 62% | 74% | 67% | 84% | 88% | 70.1% |
| *focus of emotions* | 56% | 43% | 71% | 74% | 83% | 89% | 75% | 69.4% |
| *emotions* | 64% | 65% | 76% | 68% | 79% | 47% | 63% | 66.7% |
| *eras* | 32% | 35% | 48% | 53% | 50% | 74% | 63% | 49.0% |
| *religious/spiritual views/afterlife views* | 36% | 30% | 67% | 53% | 54% | 53% | 50% | 48.3% |
| *speaker's life experiences/life events* | 0% | 0% | 10% | 11% | 17% | 26% | 25% | 11.6% |
| *post-loss actions/call to action* | 4% | 9% | 14% | 5% | 8% | 16% | 25% | 10.9% |
| *other* | 8% | 0% | 10% | 16% | 4% | 11% | 13% | 8.2% |
| **Grand Total** | 100% | 100% | 100% | 100% | 100% | 100% | 100% | 100.00% |

**Fig 9. Sample of qualitative code frequency.** Percentage of interviews (in each mode) that had a least one code applied from the given category. These code categories are from the Death question. Example interpretation: 55% of Survey interviews were tagged with at least one code related to *distance*.

Similar observations were made for other questions as well. In general, we observed that certain types of codes were less likely to appear in survey modes. However, we concluded that this effect was likely due more to the lack of followup questions than to some other quality inherent to the mode. Among modes that did afford followup questions, our exploratory analysis didn't suggest any consistent trends in code distribution. We hypothesize that if our surveys did use followup questions (e.g. using a dynamic survey method or bot interviewers [19]), we would see no differences between surveys and other modes.

**Rare codes occur more in verbal modes.** Returning to the discussion of thematic content, the aspiration of most qualitative interviewing is not to gather as many codes as possible, but rather to represent a diversity of experience. We reasoned that rare codes were thus also of interest in data equivalence. As a unit of analysis, rare codes can be interpreted as such: If an interview has a rare code, it indicates that the interview successfully elicited a unique, individual experience. Modes with more rare codes may be better at eliciting diverse individual experiences.

To define rare codes, we calculated the frequency of each code across all interviews and modes, and examined the distribution of frequencies. We defined *rarity* in two different ways. First, a code was labelled as rare if its frequency was 1 standard deviation below the mean (frequency $< 3$). Second, a code was labelled as rare if its frequency fell into the bottom quartile of the distribution (frequency $< 7$). We examined both definitions of *rare* over both the *followups removed* and *followups included* subsets of data. The full statistical results are available in S10 Table. Using the first definition of rarity, we did not observe a statistically significant difference across mode (*followups removed*: $F_{6,140} = 1.618$, $p > .05$, *followups included*: $F_{4,94} = 0.924$, $p > .05$)

Under the second definition of rarity, we found statistically significant differences in the mean number of rare codes per interview across mode. The distribution of rare codes across interviews is shown in Fig 10. The differences in mean for the *followups included* subset were substantial ($F_{4,94} = 4.437$, $p < .01$), while the differences for the *followups removed* subset were borderline significant ($F_{6,140} = 2.097$, $p \approx .05$). A Tukey comparison test for the *followups included* subset confirmed that rare codes were statistically significantly higher in the audio condition than in the chat or email conditions (both at $p < .01$, for details see S12 Table).

However, the absolute difference in rare codes is small. Audio interviews had an average of two rare codes per interview, while the other modes averaged one. We noticed a possible outlier in the set of audio interview that had six rare codes. This may have inflated the mean and overestimated the magnitude of the differences. To confirm it was not a spurious outlier, we removed that point and re-ran the ANOVA. The test confirmed that the relevant values were still significant. Observing that the video interviews also varied more than other modes, we weakly hypothesize that verbal interviews may help elicit more unique, individual experiences. A larger sample size of interviews would provide more evidence to this claim.

## Discussion

When designing an interview study, especially in an online context, there are many factors to consider. This is reflected in the breadth of our analysis. Yet despite the daunting list of considerations, we find that most are not affected by mode. Most importantly, for individuals recruited from online panels who are willing to participate in studies conducted using a wide range of modes, mode does not appear to pose a threat to the study's underlying validity. However, this result may not hold for studies where participants are recruited outside of online panels or with participants who express strong preferences against certain interview modes. In the following sections we will discuss (1) how mode does not affect the structure or thematic

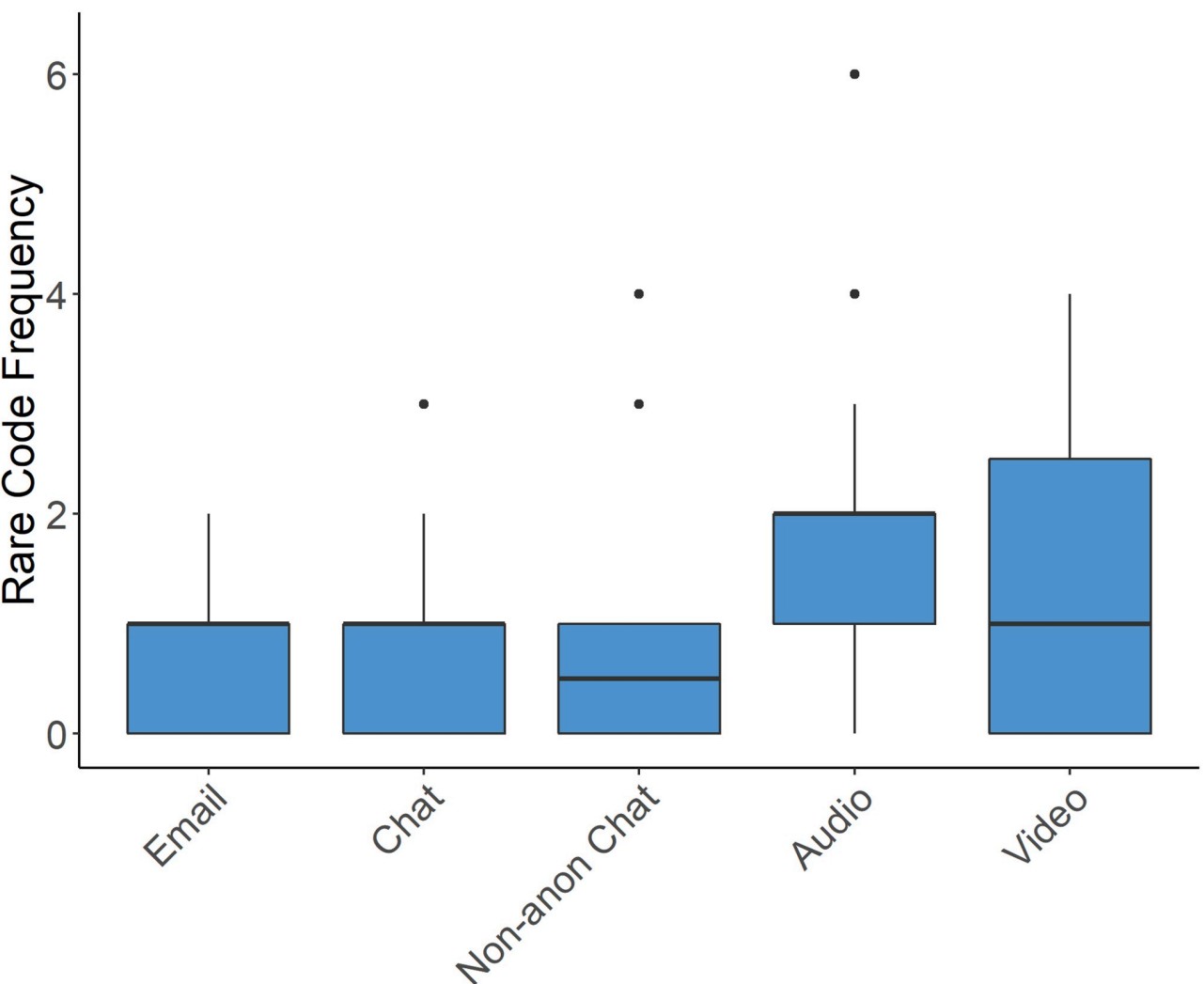

**Fig 10. Distribution of rare codes by mode.** Number of rare codes per interview per mode. A codes is *rare* if its frequency falls in the first quartile of the overall code distribution. Audio interviews were statistically significantly higher than Email and Chat interviews.

content of the data collected, (2) how logistical differences across mode raise practical questions and potentially introduce bias, (3) how mode can also be a contextual question, (4) why anonymity may be taken for granted by participants, and (5) the ethical considerations to weigh beyond the measures discussed in this study.

## Mode does not significantly impact interview data

We began this study hoping to answer the question: will my choice of online interview mode meaningfully impact my interview data? We examined a wide variety of relevant confounding variables and outcome measures, from rapport to self-disclosure to distributions of qualitative codes. **Nearly all the analyses directly related to data content, equivalence, or quality yielded null results**. In cases where we did find qualitatively significant or statistically significant results, the effect sizes were relatively low. For example, even though we designed the chat and non-anonymous chat conditions to elicit differences in perceived anonymity, the practical magnitude of the effect was surprisingly small, dropping less than one point on the

seven point scale. In another example, while we observed that audio and video conditions yielded a greater number of rare topics, the difference amounted to one or two additional rare topics per interview.

While the null results can be attributed to the low statistical power in our relatively small sample of interviews, our sample sizes, ranging from 18 to 26 interviews per mode, align with those used across a variety of academic disciplines. Therefore, we would expect that studies of a similar size are unlikely to observe statistically significant differences as well. In addition, the average effect size (Cohen's d) for the number of qualitative codes that we observe across non-survey modes, none of which were statistically significant, was 0.39 ($\sigma = 0.26$). This is generally considered to be a small to medium effect. Increasing the sample size, while possibly making these effects significant, would only further decrease the effect size [52]. Thus despite low power, **the effects of mode on interview data are likely small enough that most interviewers can cautiously ignore them**. Furthermore, evidence from previous work suggests that the effect of mode is likely even smaller for interviews that focus on less sensitive content [27]. Of course, this conclusion does not apply to large-scale collection methods like the broad distribution of a survey or questionnaire interview.

## Logistical differences across mode are important

Instead, **the most qualitatively and quantitatively significant effects of interview mode were related to logistics**. This raises two important considerations for researchers: one regarding practicality and a second regarding validity. First, in terms of practical considerations, we observe differences in recruitment, cost, and difficulty across mode. From the outset, a majority of our participant pool was unwilling to complete audio, video, and/or recorded interviews. Even among those who were willing to participate in these kinds of interviews, scheduling and completion rates aligned with the general trend in mode preferences: audio and video interviews trailed behind other modes at <40% completion. The estimated costs associated with audio, video, and chat modes were also substantially higher than survey and email. Even word count, the outcome with the largest effect sizes by mode, was arguably better interpreted as a logistical constraint, as in the cost of transcription, rather than some meaningful difference in qualitative data.

While these factors are significant for resource constraints, they are also represent important personal constraints for researchers to consider. In weekly check-in meetings during the course of this study, interviewers frequently voiced irritation with participants dropping out last minute or not showing up for interviews. For example, in a post-interview survey Interviewer 3 reported being frustrated with one of the participants who had signed up to complete an audio interview stating, "This was frustrating. The interviewee did not show for their first interview, then chose to reschedule. Then they did not show for their rescheduled interview. . ." This event, which was not an isolated occurrence, shows that matters of practical concern also have personal and emotional consequences that researchers should consider. These factors are likely even more important in the context of large-scale or longitudinal interview studies.

Second, the logistical differences across mode also raise concerns over validity and potential bias. Specifically, difficulty in recruiting participants for audio and video interviews, evidenced by the low willingness to participate and high drop out rates, may lead to self-selection bias in these modes. While the majority of potential participants had access to a webcam and keyboard, it is possible some participants did not feel comfortable using the technology. Even if we assume general comfort with the process of using a device, they may not be comfortable with having an interviewer see their face or their surroundings, hear their voice, or risk being

overheard by others who live or work in their space. While we do not have direct measurements of these effects, it is possible that these more personally revealing modes (i.e. revealing voice and appearance) may dissuade shy, self-conscious, or privacy-concerned individuals. Audio and video modes are also less conducive to multi-tasking as text-based modes. This may make these modes less attractive to participants, especially digital workers, and pose more significant scheduling challenges. Furthermore, a lack of private space, the risk of being overheard by those around them, or a limited internet connection may cause users who have access to the requisite devices to still not participate. These hurdles are likely to be even more significant for individuals with lower income.

## Mode can also be a personal, contextual question

While our statistical analysis did not provide evidence that mode affects interview data in a substantial way, the responses of several individual participants provide a different perspective. For some interviewees, mode was salient enough that they voluntarily discussed its impacts on their experience without any prompting. These comments tended to be offered from (1) those who valued anonymity, (2) those who found it easier to disclose information with less direct interviewer contact, or (3) those in chat modes, where novelty may have played a role. While these individual experiences do not reflect general trends, qualitative researchers rightfully strongly emphasize the importance of individual experiences. Additional consideration should be given to the population being studied when deciding upon interview mode.

Previous research has argued that specific details surrounding the context of an interview should be a large consideration. Salmons (2011) encourages study designers to **ask whether the interview mode is "aligned with research purpose, interview style, and access/preference of participants"** and whether the interview experience "mirrors interactions being studied" [2, p. 13,21]. In some cases, the answers to these questions are clear. For example, Stanko and Richter (2011) explored the role of identity and routine in online worlds (e.g., Second Life, World of Warcraft), and chose a virtual world environment to conduct interviews as it was organic to the topic [53]. In a study of a teen after-school writing program, Deegan chose Facebook chat and email, as the pool of teens had an already established community on Facebook [54]. Generally, researchers who want to watch their interviewees react to something visual or interact with a user interface may choose a video mode or possibly another mode augmented with video screen capture.

However, in many cases the alignment of research environment and mode is not so clear. Many communities of interest do not have clear online home communities yet. Particularly in the context of the 2020 pandemic, human subjects research has rapidly moved to online modes. However, many participant pools may have heterogeneous digital preferences or may not have a coherent virtual analogue for their community as they lack an obvious "contextual naturalness" [55]. In these circumstances, mode and communication preferences may be subject to individual factors, such as personality or personal experiences, that are inaccessible to interviewers prior to the study. Furthermore, the context of the methodological approach underpinning the study, the aim of the research, and the target population may dictate the choice of interview mode. The design of certain studies might not be amenable to all interview modes equally and specific populations may be easier to recruit using certain modes due to access, comfort, or anonymity issues. It is in these cases where we hope that this study is of help to qualitative researchers, relieving the concern that any mode generally affects interview data or its validity. As many interviewers long before us have noted, it's generally advantageous to be flexible with mode, allowing the participant to guide the decision [1, p. 44]. Our results also provide evidence that mixing interview modes within a study is unlikely to be a source of

data validity issues. Taking all of these factors into account, we conclude that **in the general case, choosing a mode is better framed as a practical, personal, or contextual question rather than a meaningful threat to the data collected**.

### Many participants trust that they are anonymous

As researchers who often work in the realm of online privacy, we took extra steps in the design of the interview process to ensure that most participants had a strong guarantee of anonymity if they chose to take it. Privacy and anonymity were valued and remarked on by many participants. However, on the whole, participants seemed to feel they were essentially anonymous regardless of condition, whether they provided their name, or showed their face. As a result, building extra logistical or infrastructural steps to protect privacy may not be necessary if done in the name of participant comfort. That said, steps for protecting participant privacy may still be advisable or even required for responsible data protection. Steps that ensure that participants' identifying information is either not collected at all or not stored with interview recordings or transcripts can reduce the chance that participants might be identified in the event that interview data is accidentally or maliciously released or subject to legal subpoena.

### Ethical considerations: Extracting guts and hearts

Whatever the methodological unpinning, at their strength qualitative interviews are collaborative endeavors that cultivate fertile ground for responses rather than simply extracting them. While we investigated outcomes such as self-disclosure, honesty, and perceived anonymity, it is likely never the case that these are outcomes to strongly optimize for. For example, respectful interview design not only considers the participant's perceived anonymity, but also thoughtfully models possible threats to privacy and confidentiality.

One Video participant said tongue-in-cheek in their post-interview survey: "I ripped into my guts and pulled out the beating heart as it was. you're welcome." We examined an abundance of outcome factors in our analysis, because simply optimizing for self-disclosure or intimacy can result in ripped guts rather than rich interaction.

## Conclusions

Audio, video, chat, email, or survey: How much does online interview mode matter? Overall, we found little evidence to suggest that any of these modes had substantial impacts on the validity or data equivalence of interviews. However, we observed that mode was more likely to impact logistical factors. In particular, substantial differences in recruitment difficulty, time, and cost were found across mode. In addition, we note some anecdotal qualitative differences between modes related to rapport, disclosure, and anonymity. While we observed that individuals had a diversity of experiences across modes, in general we conclude that when choosing between online interview modes researchers can safely de-emphasize data validity as a source of concern.

Many of our observations were likely intuitive to interview researchers. We are pleased to add a new, focused, and experimental voice that affirms much of the experiential knowledge of these researchers.

## Supporting information

**S1 Protocol. Interviewer handbook.** Handbook containing interviewer instructions, interview script, and procedures.
(PDF)

**S2 Protocol. Qualitative coding handbook.** Handbook containing qualitative coding instructions and procedures.
(PDF)

**S3 Protocol. Interviewee screening survey.** Transcript of the screening survey taken by potential participants to determine their eligibility.
(PDF)

**S4 Protocol. Interviewee post-interview survey.** Transcript of the post-interview survey taken by the interviewee after their interview.
(PDF)

**S5 Protocol. Interviewer post-interview survey.** Transcript of the post-interview survey taken by the interviewer after each interview they conducted.
(PDF)

**S1 Table. Interview scheduling rates.** ANOVA and Tukey comparison results testing differences in scheduling rates across mode.
(PDF)

**S2 Table. Interview completion rates.** ANOVA and Tukey comparison results testing differences in completion rates across mode.
(PDF)

**S3 Table. Self-reported self-disclosure.** ANOVA and Tukey comparison results testing differences in interviewees' self-reported self-disclosure across mode.
(PDF)

**S4 Table. Self-reported honesty.** ANOVA and Tukey comparison results testing differences in interviewees' self-reported honesty across mode.
(PDF)

**S5 Table. Self-reported perceived anonymity.** ANOVA and Tukey comparison results testing differences in interviewees' self-reported perceived anonymity across mode.
(PDF)

**S6 Table. Interviewee word count.** ANOVA and Tukey comparison results testing differences in interviewee word across mode.
(PDF)

**S7 Table. Interviewee word count excluding followup questions.** ANOVA and Tukey comparison results testing differences in interviewee word across mode excluding responses to followup questions.
(PDF)

**S8 Table. Qualitative codes.** ANOVA and Tukey comparison results testing differences in the frequency of qualitative codes across mode.
(PDF)

**S9 Table. Qualitative codes excluding followup questions.** ANOVA and Tukey comparison results testing differences in the frequency of qualitative codes across mode excluding responses to followup questions.
(PDF)

**S10 Table. Rare qualitative codes (standard deviation).** ANOVA and Tukey comparison results testing differences in the frequency of rare qualitative codes (two standard deviations

method) across mode.
(PDF)

**S11 Table. Rare qualitative codes (standard deviation) excluding followup questions.**
ANOVA and Tukey comparison results testing differences in the frequency of rare qualitative
codes (two standard deviations method) across mode excluding responses to followup questions.
(PDF)

**S12 Table. Rare qualitative codes (quartiles).** ANOVA and Tukey comparison results testing
differences in the frequency of rare qualitative codes (first quartile method) across mode.
(PDF)

**S13 Table. Rare qualitative codes (quartiles) excluding followup questions.** ANOVA and
Tukey comparison results testing differences in the frequency of rare qualitative codes (first
quartile method) across mode excluding responses to followup questions.
(PDF)

**S14 Table. Rare qualitative codes (quartiles) excluding outlier.** ANOVA and Tukey comparison results testing differences in the frequency of rare qualitative codes (first quartile method)
across mode. One outlier was removed to test its effect.
(PDF)

**S15 Table. Rare qualitative codes (quartiles) excluding outlier and followup questions.**
ANOVA and Tukey comparison results testing differences in the frequency of rare qualitative
codes (first quartile method) across mode excluding responses to followup questions. One outlier was removed to test its effect.
(PDF)

**S16 Table. Willingness to participate.** Multiple t-test results comparing screening survey
results pre-pandemic and during the pandemic including the availability of remote interview
resources, accessibility of a private space, and willingness to participate in various interview
modes.
(PDF)

## Acknowledgments

Many thanks to our interviewer team for their dedication and flexibility. Gratitude to all the
interviewees for their openness and generosity. Soraya Alli and Era Vuksani both crucially
helped frame the study in its early days.

## Author Contributions

**Conceptualization:** Maggie Oates, Kyle Crichton, Lorrie Cranor.

**Data curation:** Maggie Oates, Kyle Crichton, Storm Budwig, Erica J. L. Weston, Brigette M.
   Bernagozzi, Julie Pagaduan.

**Formal analysis:** Maggie Oates, Kyle Crichton, Storm Budwig, Erica J. L. Weston, Brigette M.
   Bernagozzi, Julie Pagaduan.

**Funding acquisition:** Lorrie Cranor.

**Investigation:** Maggie Oates, Kyle Crichton, Storm Budwig, Erica J. L. Weston, Brigette M.
   Bernagozzi, Julie Pagaduan.

**Methodology:** Maggie Oates, Kyle Crichton.

**Project administration:** Maggie Oates, Kyle Crichton, Lorrie Cranor.

**Resources:** Lorrie Cranor.

**Software:** Maggie Oates, Kyle Crichton.

**Supervision:** Maggie Oates, Kyle Crichton, Lorrie Cranor.

**Validation:** Maggie Oates, Kyle Crichton, Lorrie Cranor.

**Visualization:** Maggie Oates, Kyle Crichton.

**Writing – original draft:** Maggie Oates, Kyle Crichton, Lorrie Cranor.

**Writing – review & editing:** Maggie Oates, Kyle Crichton, Lorrie Cranor.

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
