## [Decision Letter · Decision Letter 0]

23 Jun 2021

PONE-D-21-13856

Audio, video, chat, email, or survey: How much does online interview mode matter?

PLOS ONE

Dear Dr. Crichton,

Thank you for submitting your manuscript to PLOS ONE. After careful consideration, we feel that it has merit but does not fully meet PLOS ONE’s publication criteria as it currently stands. Therefore, we invite you to submit a revised version of the manuscript that addresses the points raised during the review process.

We look forward to receiving your revised manuscript.

Kind regards,

Janet E Rosenbaum, Ph.D.

Academic Editor

PLOS ONE

Journal Requirements:

a) If there are ethical or legal restrictions on sharing a de-identified data set, please explain them in detail (e.g., data contain potentially sensitive information, data are owned by a third-party organization, etc.) and who has imposed them (e.g., an ethics committee). Please also provide contact information for a data access committee, ethics committee, or other institutional body to which data requests may be sent

Additional Editor Comments (if provided):

Reviewers' comments:

Reviewer's Responses to Questions

**Comments to the Author**

1. Is the manuscript technically sound, and do the data support the conclusions?

Reviewer #1: Partly

Reviewer #2: Yes

2. Has the statistical analysis been performed appropriately and rigorously? 

Reviewer #1: Yes

Reviewer #2: I Don't Know

3. Have the authors made all data underlying the findings in their manuscript fully available?

Reviewer #1: Yes

Reviewer #2: Yes

4. Is the manuscript presented in an intelligible fashion and written in standard English?

Reviewer #1: Yes

Reviewer #2: Yes

5. Review Comments to the Author

Reviewer #1: The authors should be commended for implementing an experimental design to assess key differences among major methods of online interviewing. The authors also address the important features of interviews related to data quality, including rapport with the interviewer, honesty, and self-disclosure. This experiment and the results are a useful contribution to the literature for practitioners, including the relative cost data presented in Table 5.

At the same time, there are a number of limitations to this research which deserve more attention from the authors:

The main questions the authors test are clearly designed to represent questions with different degrees of sensitivity. These questions do not seem like questions that would be asked by survey or market researchers conducting online interviews. Including some “standard” survey questions would have provided readers with more confidence that these results are relevant to their own research. Can the authors comment on how their selection of questions could have affected the results? The interviewee post-interview survey was a useful design feature, but responses to some of these questions - desirable responding, self-disclosure, and honesty - could have been directly influenced by the specific questions in the initial interview.

The sample size is sufficient for most analysis, but not all, as the authors note on page 23. For example, an analysis of interviewer effects showing interviewers’ contribution to variance in participants responses to questions. The authors note on page 16 that they examined interview completion rates, mode completion rates, technical difficulties, and rapport rating and did not see any differences. These are useful metrics to report, but do not actually identify interview effects. If the authors cannot estimate interviewers’ contribution to variance due to sample size or study design, are there any further indirect metrics that could inform the likely potential for interviewer effects?

On page 27, the authors note that 99% of screened participants indicated having access to a computer with a keyboard. Part c. of Table 4 (on page 26) also shows participants were disproportionately white. Like most examples of online research I have seen, this sample is clearly skewed to higher average SES and White participants. Although it seems likely that the authors could not have obtained a more representative sample without significantly changing recruitment methods, it would be useful for the authors to comment on how this skew in participant demographics might have affected results across all experimental conditions?

The authors note on page 24 that the greatest challenges to recruitment and logistics involved audio and video modes. They specifically note on page 26 that unwillingness to participate in video mode was the greatest disqualifying factor in the recruited pool. This is not surprising, but good to note for practitioners who are considering different online modes. A separate question is how the lower screening and completion rates and the higher no-show rates for audio and video modes (as reported in Table 5 on page 32). How might this differential nonresponse pattern have affected findings and conclusions?

The authors’ first conclusion that mode is only a practical question, and not a threat to validity, is not supported by relevant literature on modes and accuracy of response and also not supported by two of their findings. First, as the authors note, the greatest challenge to recruitment (and logistics) involved audio and video mode. Given that 99% of screened participants indicated having access to a computer with a keyboard and 70% having a webcam, it seems like unwillingness to participate in audio or video modes was influenced (at least in part) by recruits’ lack of comfort with reporting in those modes. Audio and video do complicate logistics, as the authors note, but having these devices suggests these participants should be comfortable using them. Even though the authors did not find significant results across modes in their analysis of disclosure, they note on page 36 that interviewers might overestimate participants’ comfort and on page 42 that some participants indicated mode affected their disclosure. So, this first conclusion seems stronger than the research demonstrates, especially considering the noted limitation in sample size that affected all quantitative analysis.

Reviewer #2: Dear Kyle Crichton and Co-authors,

Many thanks for the opportunity to review your manuscript, “Audio, video, chat, email or survey: How much does interview mode matter?”

You have presented a very well-written and clear manuscript. You aptly highlighted the need for the novel research and detailed the methods, and findings meticulously. Although I thoroughly enjoyed reading it for its clarity and novelty, I must admit that it was challenging for me at times as a qualitative researcher. The statement on page 11, line 281 (“we were wary of attempting to assess a qualitative method using quantitative …”), was a reassuring touch.

I do have some feedback and suggestions for your consideration. Some quite minor and some that may require more thought. Please note, that I have reviewed the manuscript from a qualitative researcher perspective and have not thoroughly reviewed the statistical analyses.

The study title and aims suggest that the research compares empirically, different modes of online interviews, and yet you have included surveys. You note that surveys are “the interview’s close cousins” however, qualitative researchers would likely argue that they are a different data collection technique to interviews altogether (i.e., rather than a mode of online interview). The rationale for including survey techniques in the study is not overly satisfying and it seems that more needs to be said about the apparent juxtaposition of online interview modes and survey techniques.

Differences between qualitative methodologies is not addressed. For instance, a phenomenological study would not be amenable to data collection via a survey, nor would coding of data be a consideration. Ethnographic studies also typically rely on observational data, and this is not mentioned at all. You duly acknowledge that there is no universal approach to qualitative data analysis, however you have not stated that the approach depends on the aims of the research and the methodology (consider narrative analysis, discourse analysis, etc.). It seems that the findings of the study are applicable to mixed methods research, but not so much to purely qualitative research (although this clearly depends on the methodology). Perhaps this could be considered and discussed.

On page 48, code count is discussed as a measure of thematic analysis potential. You acknowledge that this is simplistic, a point with which I agree, but I also challenge this as a meaningful measure of the qualitative equivalence. To my mind, the number of codes (i.e., ways to break up and categorise data) does not speak to its interpretative potential. At minimum, a rationale for including this as a measure with an appropriate reference would be useful.

In terms of minor feedback, it would be useful to know if any qualitative data analysis software or tools was used by the coders. On page 3 lines 44-45 need revision. On page 4 line 83, there is a direct quote attributed to source 8, but no page number. As very minor feedback, please check all direct quotes - some of the quotation marks are orientated the wrong way. On page 23, line 603, it says that 145 interviews were completed; elsewhere it says 154.

I was surprised to see you refer to the limitations in the method section, however, I recognise this may be a more accepted convention in quantitative research papers. It was pleasing to see that you conducted a small-scale screening study to measure the impact of the COVID-19 pandemic on willingness to participate in the various modes.

With many thanks once again, I sincerely look forward to seeing this manuscript again and hopefully eventually in print.

Very best,

Reviewer

6. PLOS authors have the option to publish the peer review history of their article (what does this mean?). If published, this will include your full peer review and any attached files.

Reviewer #1: No

Reviewer #2: No

---

## [Author Response · Author response to Decision Letter 0]

26 Jul 2021

Our response to the reviewers' and editors' comments were submitted in the response to comments document and the revised cover letter. The content of the response to comments document has been copied below for reference.

We appreciate the reviewers’ constructive feedback and have incorporated changes based on their comments into the revised version of the manuscript. Our response is organized as follows: first we cover the most significant comments that were primarily related to the interpretation of results, then we move on to address the issues raised regarding our methodology, and finally we conclude with comments related to potential bias stemming from the study design. In addition to the comments addressed below, several minor revisions were made to address grammatical errors, issues with direct quotes, and a discrepancy in the total number of interviews reported (154) noted by the reviewers.

Comments Related to Discussion and Interpretation

Comment (Reviewer 1): The authors’ first conclusion that mode is only a practical question, and not a threat to validity, is not supported by relevant literature on modes and accuracy of response and also not supported by two of their findings. First, as the authors note, the greatest challenge to recruitment (and logistics) involved audio and video mode. Given that 99% of screened participants indicated having access to a computer with a keyboard and 70% having a webcam, it seems like unwillingness to participate in audio or video modes was influenced (at least in part) by recruits’ lack of comfort with reporting in those modes. Audio and video do complicate logistics, as the authors note, but having these devices suggests these participants should be comfortable using them. Even though the authors did not find significant results across modes in their analysis of disclosure, they note on page 36 that interviewers might overestimate participants’ comfort and on page 42 that some participants indicated mode affected their disclosure. So, this first conclusion seems stronger than the research demonstrates, especially considering the noted limitation in sample size that affected all quantitative analysis.

Response: To start with the most considerable modification, we agree with Reviewer 1 that our first conclusion regarding mode as a practical question was too strong and required further qualification. While the primary differences observed across mode were related to logistics, and these differences have practical implications for researchers, we agree that mode is not only a practical decision and there are valid concerns about bias and validity pertaining to mode. As such, we reframed this section to discuss the practical considerations that mode raises as well as potential bias related to self-selection (where more personally revealing modes may dissuade shy, self-conscious, or private individuals) and systematic exclusion (where lack of access to a private space or a good internet connection might preclude participation). Therefore, we modified our conclusions to (1) highlight the lack of differences in the structure and thematic content of the data collected across mode, (2) identify the logistical challenges related to practicality and validity, and (3) discuss other contextual factors that are important when deciding upon interview mode. 

Comment (Reviewer 2): Differences between qualitative methodologies is not addressed. For instance, a phenomenological study would not be amenable to data collection via a survey, nor would coding of data be a consideration. Ethnographic studies also typically rely on observational data, and this is not mentioned at all. You duly acknowledge that there is no universal approach to qualitative data analysis, however you have not stated that the approach depends on the aims of the research and the methodology (consider narrative analysis, discourse analysis, etc.). It seems that the findings of the study are applicable to mixed methods research, but not so much to purely qualitative research (although this clearly depends on the methodology). Perhaps this could be considered and discussed.

Response: As a part of the contextual elements that influence the choice of interview mode (point 3 from above), we agree with Reviewer 2 that we were missing a discussion of how the qualitative methodology and underlying aim of the research affect the decision. We incorporated this aspect into the discussion section and further restricted the applicability of our results to qualitative interview studies specifically rather than qualitative research more broadly.

Comment (Reviewer 1): On page 27, the authors note that 99% of screened participants indicated having access to a computer with a keyboard. Part c. of Table 4 (on page 26) also shows participants were disproportionately white. Like most examples of online research I have seen, this sample is clearly skewed to higher average SES and White participants. Although it seems likely that the authors could not have obtained a more representative sample without significantly changing recruitment methods, it would be useful for the authors to comment on how this skew in participant demographics might have affected results across all experimental conditions?

Response: As a part of the above changes, we also addressed two of Reviewer 1’s comments related to bias in our recruited sample. First, as Reviewer 1 correctly notes, our sample overrepresents white participants and, while we did not collect socioeconomic data directly, exhibits a skew associated with higher income. When reporting our sample’s demographics, we added a discussion of this overarching bias in the sample and in our discussion we limited the generalizability of our results to samples recruited through online panels which are likely to encounter similar bias. In addition, we discuss how access to technology, private space, and a good internet connection, all larger hurdles for individuals with lower income, can affect both recruitment in online studies and a researcher’s decision to use a certain mode. However, we also note that while we observe this general skew in race and other socioeconomic indicators, these factors were distributed relatively equally across mode and therefore are unlikely to have affected our comparison of results between conditions. 

Comment (Reviewer 1): The authors note on page 24 that the greatest challenges to recruitment and logistics involved audio and video modes. They specifically note on page 26 that unwillingness to participate in video mode was the greatest disqualifying factor in the recruited pool. This is not surprising, but good to note for practitioners who are considering different online modes. A separate question is how the lower screening and completion rates and the higher no-show rates for audio and video modes (as reported in Table 5 on page 32). How might this differential nonresponse pattern have affected findings and conclusions?

Response: As for Reviewer 1’s second comment regarding bias, we agree that differences in eligibility, completion, and no-show rates for audio and video do affect our conclusions. As previously discussed, we added several sections to the findings and discussion sections that highlight the potential self-selection bias that these modes introduce. We reason that while users may be comfortable with the process of using a device, they may not be comfortable with having an interviewer see their face or their surroundings, hear their voice, or perhaps be overheard by others who live or work in their space. Thus discomfort is not limited to knowledge of using a device. Furthermore, participants may not be interested in participating in a study with video or audio because it would require them to participate in a quiet space with a good internet connection and may be less conducive to multi-tasking.

Comments Related to the Methodology

Comment (Reviewer 2): You have presented a very well-written and clear manuscript. You aptly highlighted the need for the novel research and detailed the methods, and findings meticulously. Although I thoroughly enjoyed reading it for its clarity and novelty, I must admit that it was challenging for me at times as a qualitative researcher. The statement on page 11, line 281 (“we were wary of attempting to assess a qualitative method using quantitative …”), was a reassuring touch.

Response: In terms of our methods, we certainly agree with the skepticism expressed by Reviewer 2 and appreciate their acknowledgement of our similar hesitancy to apply qualitative analysis to evaluate quantitative research methods. This was part of the reason why we chose to move our discussion of the limitations and validity of our study further up in the paper (in the methods section) rather than the more typical location as a part of the discussion which Reviewer 2 noted. In doing so, we hoped to express these considerations up front and provide additional context before presenting our findings. 

Comment (Reviewer 2): I was surprised to see you refer to the limitations in the method section, however, I recognise this may be a more accepted convention in quantitative research papers.

Response: As discussed above, our aim in moving the limitations into the methods section, rather than the discussion, was to highlight limitations and the context in which our results should be interpreted before diving into them.

Comment (Reviewer 2): On page 48, code count is discussed as a measure of thematic analysis potential. You acknowledge that this is simplistic, a point with which I agree, but I also challenge this as a meaningful measure of the qualitative equivalence. To my mind, the number of codes (i.e., ways to break up and categorise data) does not speak to its interpretative potential. At minimum, a rationale for including this as a measure with an appropriate reference would be useful.

Response: We also agree with Reviewer 2 that code count is not a strong indicator of thematic analysis potential and restricted our interpretation to what it should have been to start: a proxy for the breadth of content disclosed during the interview. In the methods section we added our reasoning for using this outcome measure, namely that the ``generation of themes via coding... and categorization... is arguably the most common analytic approach taken by qualitative researchers using interviews'' (Roulston 2014 in SAGE Handbook of Qualitative Data Analysis). As a common methodology in qualitative research, we propose that differences in the frequency and distribution of codes across mode would be more insightful than assessing differences in word count alone.

Comment (Reviewer 2): It would be useful to know if any qualitative data analysis software or tools was used by the coders.

Response: Related to the qualitative coding, we added clarification to the methods section to specify that the qualitative coding did not use any specific qualitative research software, only shared spreadsheet templates created by the research team. 

Comment (Reviewer 2): The study title and aims suggest that the research compares empirically, different modes of online interviews, and yet you have included surveys. You note that surveys are “the interview’s close cousins” however, qualitative researchers would likely argue that they are a different data collection technique to interviews altogether (i.e., rather than a mode of online interview). The rationale for including survey techniques in the study is not overly satisfying and it seems that more needs to be said about the apparent juxtaposition of online interview modes and survey techniques.

Response: We agree with Reviewer 2 that surveys and interviews are inherently distinct methodologies. While we treated the two very differently in our quantitative analysis, having two sets of statistical tests (one with surveys and one without) for most outcome measures, we did not present the two as such in our introduction, related work, or methods sections. As such, we modified our presentation of the two methods to highlight their differences, specifically removing the sentence the reviewer referred to in their comment. In addition, we added further justification for our inclusion of the survey conditions which were to be used as a baseline for comparison and as a means of evaluating the effects of scheduling-- something that would have been very difficult and expensive to do using interview methods alone.

Comments Related to Potential Bias

Comment (Reviewer 1): For example, an analysis of interviewer effects showing interviewers’ contribution to variance in participants responses to questions. The authors note on page 16 that they examined interview completion rates, mode completion rates, technical difficulties, and rapport rating and did not see any differences. These are useful metrics to report, but do not actually identify interview effects. If the authors cannot estimate interviewers’ contribution to variance due to sample size or study design, are there any further indirect metrics that could inform the likely potential for interviewer effects?

Response: Reviewer 1 asked about additional metrics that might be able to better identify potential interviewer effects. We agree that measuring interviewer effects directly is difficult and had attempted to minimize this possibility using three interviewers with similar traits: “all three were white, women, aged 20-25 years, fluent English-speakers, willing to broach sensitive topics, with a similar gender presentation and hairstyles, and who had no previous experience with research interviewing” (page 15). In addition to evidence that completion rates, technical difficulties, and rapport rating did not vary by interviewer, we also added that the honesty self-report, word count, and qualitative code count did not substantially vary by interviewer either. 

Comment (Reviewer 1): The main questions the authors test are clearly designed to represent questions with different degrees of sensitivity. These questions do not seem like questions that would be asked by survey or market researchers conducting online interviews. Including some “standard” survey questions would have provided readers with more confidence that these results are relevant to their own research. Can the authors comment on how their selection of questions could have affected the results?

Response: Reviewer 1 noted that the choice of interview questions, specifically their sensitive nature, might provide different results than standard survey or interview questions. We agree with this point as it was an explicit design decision. According to Nandi and Platt (2017), differences in self-disclosure, perceived anonymity, and honesty should be more pronounced in interviews focusing on sensitive topics (discussed on page 9 and 14). Therefore, we reasoned that if we were to find differences in mode it would be in these kinds of sensitive interviews and accordingly we would expect to see an even smaller effect in interviews with less sensitive and more standard content. We elaborated on this in the methods section as well as made it a specific point to probe in the discussion section. 

Comment (Reviewer 1): The interviewee post-interview survey was a useful design feature, but responses to some of these questions - desirable responding, self-disclosure, and honesty - could have been directly influenced by the specific questions in the initial interview.

Response: Reviewer 1 raised concern that the choice of interview questions might bias the responses in the post-interview survey. We agree that measures that the reviewer identifies (social desirability, honesty, and self-disclosure) could be affected by the sensitive nature of the interview questions. However, since (1) our aim was to identify differences across mode, (2) the questions were the same across all conditions, and (3) we would not expect the questions to have a differential effect across mode, we do not believe this affected our results.

---

## [Decision Letter · Decision Letter 1]

19 Dec 2021

PONE-D-21-13856R1Audio, video, chat, email, or survey: How much does online interview mode matter?PLOS ONE

Dear Dr. Crichton,

Thank you for submitting your manuscript to PLOS ONE. After careful consideration, we feel that it has merit but does not fully meet PLOS ONE’s publication criteria as it currently stands. Therefore, we invite you to submit a revised version of the manuscript that addresses the points raised during the review process.

We look forward to receiving your revised manuscript.

Kind regards,

Janet E Rosenbaum, Ph.D.

Academic Editor

PLOS ONE

Journal Requirements:

Reviewers' comments:

Reviewer's Responses to Questions

**Comments to the Author**

1. If the authors have adequately addressed your comments raised in a previous round of review and you feel that this manuscript is now acceptable for publication, you may indicate that here to bypass the “Comments to the Author” section, enter your conflict of interest statement in the “Confidential to Editor” section, and submit your "Accept" recommendation.

Reviewer #1: (No Response)

Reviewer #2: (No Response)

2. Is the manuscript technically sound, and do the data support the conclusions?

Reviewer #1: Yes

Reviewer #2: Yes

3. Has the statistical analysis been performed appropriately and rigorously? 

Reviewer #1: Yes

Reviewer #2: Yes

4. Have the authors made all data underlying the findings in their manuscript fully available?

Reviewer #1: Yes

Reviewer #2: Yes

5. Is the manuscript presented in an intelligible fashion and written in standard English?

Reviewer #1: Yes

Reviewer #2: Yes

6. Review Comments to the Author

Reviewer #1: The authors’ responses to my comments are generally sound and appreciated, I have a few follow-up points for the authors to consider.

In response to my comment about how the observed skew in participant demographics to white and higher SES participants might have affected results (across all experimental conditions, the authors note that these factors were distributed relatively equally across mode and therefore are unlikely to have affected our comparison of results between conditions. The authors’ then conclude that this demographic skew should not have affected the internal validity of their experimental study, given the similar composition across conditions. To clarify my original question, a number of the authors’ conclusions include claims of external validity – that is, what other researchers should expect in terms of data quality when using different modes of online interviewing. Can the authors address the issue of how the demographic skew of their study participants could have affected the ability of their study to provide external validity?

The authors note on page 51 that “despite low power, any effects on interview data that was missed in this study are likely small enough that most interviewers can gently ignore them.” First, I assume the authors meant to say “generally” instead of gently. Second, this statement feels a bit strong, given that the authors have no statistical criteria to determine whether observed effects should, or should not, be considered meaningful. The authors’ conclusion might be correct (and I would guess it is, based on their results), but assuming this is the case without statistical support or other metrics seems questionable. Do the authors have other evidence or literature they can cite to support this conclusion that the effects are likely small enough to be ignorable?

Further in the same paragraph, the authors conclude that they “... expect the effect of mode to be even smaller for interviews that focus on less sensitive content.” This statement makes intuitive sense to me but given that the authors (intentionally) did not include less sensitive questions in the experimental design or cite relevant literature here, the basis of this statement needs to be more clearly justified.

Reviewer #2: Dear Kyle Crichton and Co-authors,

Many thanks for the opportunity to review your revised manuscript, “Audio, video, chat, email or survey: How much does interview mode matter?”

Thank you for the changes you have made in response to reviewer feedback. You have addressed a range of critical comments thoughtfully.

You have gone some way in addressing the key criticism made about methodology versus data collection method however, I do still feel that there is some conflation of qualitative research and interviews. Specifically, on p. 5, under the sub-heading “Honesty”, you state, “In qualitative research, the honesty of a participant’s response is a fundamental underpinning of the field.” I would argue this is also the case in quantitative research, where self-report measures are used (e.g., validated pain scales, perceived knowledge of a particular topic, satisfaction surveys, etc.). Equally, ethnographic studies, for example, data is collected in part by researchers’ observations of participants (no honesty required on participants’ part). I feel that some further clarity (a sentence or two) about the fact that you are not investigating qualitative research methods generally, but rather a data collection method commonly used in qualitative research, namely interviews.

There are some other minor points for your consideration on p. 37 (lines 965-966) the terms interviewee and participant are used interchangeably which is a bit confusing. There is a minor typographic error on p.7 (line 157) “interviewees”.

Thank you again for your thorough revision and very best wishes.

7. PLOS authors have the option to publish the peer review history of their article (what does this mean?). If published, this will include your full peer review and any attached files.

Reviewer #1: No

Reviewer #2: No

---

## [Author Response · Author response to Decision Letter 1]

13 Jan 2022

Copied from the response to reviewers document:

Thank you to the reviewers for taking the time to provide additional comments and to clarify some of the points that we had not fully addressed, or had misunderstood, in the initial revision. Below we respond to each of the reviewer’s comments individually and include a summary of the changes we made in the manuscript to address each point.

Comment (Reviewer 1): In response to my comment about how the observed skew in participant demographics to white and higher SES participants might have affected results (across all experimental conditions, the authors note that these factors were distributed relatively equally across mode and therefore are unlikely to have affected our comparison of results between conditions. The authors’ then conclude that this demographic skew should not have affected the internal validity of their experimental study, given the similar composition across conditions. To clarify my original question, a number of the authors’ conclusions include claims of external validity – that is, what other researchers should expect in terms of data quality when using different modes of online interviewing. Can the authors address the issue of how the demographic skew of their study participants could have affected the ability of their study to provide external validity?

Response: We realize that we had misinterpreted the reviewer’s initial comments and agree that the demographic skew in our data does pose a challenge to external validity that we had not adequately acknowledged in the paper. We made two changes accordingly. First, we added several lines to the “Participant demographics” sub-section (lines 708-719 in the tracked changes document) that explore the differences between interviews with black participants and those with Latino and white participants. We use this data to indicate that there are possible differences that should make researchers cautious in applying our findings, but we also state that we cannot draw specific conclusions due to the low number of black and Latino participants in each condition (about 1-2 participants in each group per arm). This prevented any meaningful subgroup analysis. Second, we expanded on our discussion of socioeconomic skew and disproportionate access to technology in the following section (lines 755-759 in the tracked changes document) to be explicit about challenges to internal versus external validity. We discuss how our sample is skewed from the general population but also note that it is a suboptimal reality that, by the same token, our sample is likely representative of online research panels. Therefore, we conclude that researchers using similar methods of recruitment can likely apply our findings directly but other researchers, particularly those working with low-income or marginalized groups, should be tentative in drawing on the same conclusions.

Comment (Reviewer 1): The authors note on page 51 that “despite low power, any effects on interview data that was missed in this study are likely small enough that most interviewers can gently ignore them.” First, I assume the authors meant to say “generally” instead of gently. Second, this statement feels a bit strong, given that the authors have no statistical criteria to determine whether observed effects should, or should not, be considered meaningful. The authors’ conclusion might be correct (and I would guess it is, based on their results), but assuming this is the case without statistical support or other metrics seems questionable. Do the authors have other evidence or literature they can cite to support this conclusion that the effects are likely small enough to be ignorable?

Response: Our original intention in using the word “gently” was to convey that the effects could be ignored with caution but shouldn’t be ignored completely. However, we agree that the word “gently” is confusing so we changed it to use the word “cautiously” to make this point more clearly. We, presumably like the reviewer, feel that “generally” would make this statement too strong and in general felt that the language we use should be softened. It now reads “Thus despite low power, the effects of mode on interview data are likely small enough that most interviewers can cautiously ignore them.” In addition to the wording, we felt that the reviewer’s criticism regarding the lack of supporting evidence in this section was a fair point. We believe that our statistically insignificant results across most of our primary metrics justify the intent behind our claim, but to make a stronger case we expanded upon our findings in this section (lines 1412-1424 in the tracked changes document). Specifically we point to the small to moderate effect sizes that we observe in the data measured using Cohon’s d (with citation) and discuss how larger sample sizes, while possibly making these effects statistically significant, would only decrease the relative effect size.

Comment (Reviewer 1): Further in the same paragraph, the authors conclude that they “... expect the effect of mode to be even smaller for interviews that focus on less sensitive content.” This statement makes intuitive sense to me but given that the authors (intentionally) did not include less sensitive questions in the experimental design or cite relevant literature here, the basis of this statement needs to be more clearly justified.

Response: In the referenced paragraph we were missing the relevant citation from Nandi and Platt (2017) that had been used in a similar discussion of sensitive interview questions in the related works section. We reworded this sentence to make it clear that the justification for this claim draws from previous work rather than our own findings. It now reads “Furthermore, evidence from previous work suggests that the effect of mode is likely even smaller for interviews that focus on less sensitive content [36]”. In reviewing this claim, we felt that an explicit reference and discussion should also be included in the related works sub-section entitled “Sensitive content” so several lines were added there (lines 157-161 in the tracked changes document).

Comment (Reviewer 2): You have gone some way in addressing the key criticism made about methodology versus data collection method however, I do still feel that there is some conflation of qualitative research and interviews. Specifically, on p. 5, under the sub-heading “Honesty”, you state, “In qualitative research, the honesty of a participant’s response is a fundamental underpinning of the field.” I would argue this is also the case in quantitative research, where self-report measures are used (e.g., validated pain scales, perceived knowledge of a particular topic, satisfaction surveys, etc.). Equally, ethnographic studies, for example, data is collected in part by researchers’ observations of participants (no honesty required on participants’ part). I feel that some further clarity (a sentence or two) about the fact that you are not investigating qualitative research methods generally, but rather a data collection method commonly used in qualitative research, namely interviews.

Response: We appreciate that the reviewer acknowledged our changes in response to their original comments and agree that a few further changes are necessary. First, in the referenced section, the reviewer is correct that honesty is an important underpinning of both qualitative and quantitative research, particularly those that rely on self-reported measures. Therefore, we shifted the start of the discussion in the “Honesty” sub-section to focus on human subjects research rather than qualitative methods specifically (lines 117-122 in the tracked changes document). In particular we acknowledge that both qualitative and quantitative researchers in this area face the challenge of accurately capturing how people behave and think in the real world. Therefore, honesty in self-reported measures are equally important, and challenging, in both contexts. Second, to clearly distinguish between research on interview methods (our work) and that on qualitative research methods, we added a paragraph to the end of the introduction where we introduce our study and its contributions (lines 64-71 in the tracked changes document). In this section we explicitly state that our research focuses on online interviews as a data collection method, the interpretation of our findings are limited to that context, and we do not attempt to expand our conclusions beyond that to qualitative methods more broadly.

Comment (Reviewer 2): There are some other minor points for your consideration on p. 37 (lines 965-966) the terms interviewee and participant are used interchangeably which is a bit confusing.

Response: We changed the referenced section to use the term “interviewee” exclusively to avoid confusion. We felt like this point warranted further examination beyond just this section so we changed “participant” to “interviewee” in several other findings sections where it made the discussion clearer. In general, we standardized the terminology to use “interviewee” when discussing findings pertaining to the content of the interviews or self-reported measures from the surveys. Special attention was paid to cases where there was a discussion of both interviewees’ and interviewers’ experiences in the same section. 

Comment (Reviewer 2): There is a minor typographic error on p.7 (line 157) “interviewees”.

Response: This was changed from “Social desirability bias is a phenomenon where interviewee’s…” to “Social desirability bias is a phenomenon where interviewees…”

---

## [Decision Letter · Decision Letter 2]

31 Jan 2022

Audio, video, chat, email, or survey: How much does online interview mode matter?

PONE-D-21-13856R2

Dear Dr. Crichton,

We’re pleased to inform you that your manuscript has been judged scientifically suitable for publication and will be formally accepted for publication once it meets all outstanding technical requirements.

Kind regards,

Janet E Rosenbaum, Ph.D.

Academic Editor

PLOS ONE

Additional Editor Comments (optional):

Reviewers' comments:

Reviewer's Responses to Questions

**Comments to the Author**

1. If the authors have adequately addressed your comments raised in a previous round of review and you feel that this manuscript is now acceptable for publication, you may indicate that here to bypass the “Comments to the Author” section, enter your conflict of interest statement in the “Confidential to Editor” section, and submit your "Accept" recommendation.

Reviewer #1: All comments have been addressed

Reviewer #2: All comments have been addressed

2. Is the manuscript technically sound, and do the data support the conclusions?

Reviewer #1: Yes

Reviewer #2: Yes

3. Has the statistical analysis been performed appropriately and rigorously? 

Reviewer #1: Yes

Reviewer #2: Yes

4. Have the authors made all data underlying the findings in their manuscript fully available?

Reviewer #1: Yes

Reviewer #2: Yes

5. Is the manuscript presented in an intelligible fashion and written in standard English?

Reviewer #1: Yes

Reviewer #2: Yes

6. Review Comments to the Author

Reviewer #1: I have no further questions, comments, or suggestions for the author. I am satisfied that the revised version addresses my major concerns from prior reviews..

Reviewer #2: Dear Kyle Crichton and Co-authors,

Many thanks for the opportunity to review your revised manuscript, “Audio, video, chat, email or survey: How much does interview mode matter?” once again.

Thank you for kindly making the suggested changes to the manuscript, as well as the additional edits and alterations. I am satisfied with the changes and the manuscript in its current state.

Wishing you all the very best.

7. PLOS authors have the option to publish the peer review history of their article (what does this mean?). If published, this will include your full peer review and any attached files.

Reviewer #1: No

Reviewer #2: No

---

## [Editor Report · Acceptance letter]

9 Feb 2022

PONE-D-21-13856R2 

Audio, video, chat, email, or survey: How much does online interview mode matter? 

Dear Dr. Crichton:

I'm pleased to inform you that your manuscript has been deemed suitable for publication in PLOS ONE. Congratulations! Your manuscript is now with our production department. 

Kind regards, 

on behalf of

Dr. Janet E Rosenbaum 

Academic Editor

PLOS ONE